biomechanics/biomedical engineering/
computational biology

biomechanics, mathematical model, locomotion, fall risk, reachability, feedback control

**Author for correspondence:**
Patrick D. Holmes
e-mail: pdholmes@umich.edu

# Characterizing the limits of human stability during motion: perturbative experiment validates a model-based approach for the Sit-to-Stand task

Patrick D. Holmes[1], Shannon M. Danforth[1],
Xiao-Yu Fu[1,3], Talia Y. Moore[2] and Ram Vasudevan[1,2]

[1]Department of Mechanical Engineering, and [2]Robotics Institute, University of Michigan, Ann Arbor, MI, USA
[3]Faculty of Kinesiology, University of Calgary, Calgary, Alberta, Canada

PDH, 0000-0001-5252-4664; SMD, 0000-0002-5825-587X;
TYM, 0000-0003-0867-4512; RV, 0000-0003-1978-0572

Falls affect a growing number of the population each year. Clinical methods to assess fall risk usually evaluate the performance of specific motions such as balancing or Sit-to-Stand. Unfortunately, these techniques have been shown to have poor predictive power, and are unable to identify the portions of motion that are most unstable. To this end, it may be useful to identify the set of body configurations that can accomplish a task under a specified control strategy. The resulting strategy-specific boundary between stable and unstable motion could be used to identify individuals at risk of falling. The recently proposed Stability Basin is defined as the set of configurations through time that do not lead to failure for an individual under their chosen control strategy. This paper presents a novel method to compute the Stability Basin and the first experimental validation of the Stability Basin with a perturbative Sit-to-Stand experiment involving forwards or backwards pulls from a motor-driven cable with 11 subjects. The individually-constructed Stability Basins are used to identify when a trial fails, i.e. when an individual must switch from their chosen control strategy (indicated by a step or sit) to recover from a perturbation. The constructed Stability Basins correctly predict the outcome of trials where failure was observed with over 90% accuracy, and correctly predict the outcome of successful trials with over 95% accuracy. The Stability Basin was compared to three other methods and was

found to estimate the stable region with over 45% more accuracy in all cases. This study demonstrates that Stability Basins offer a novel model-based approach for quantifying stability during motion, which could be used in physical therapy for individuals at risk of falling.

# 1. Introduction

Falls are the leading cause of injury in people over 75 years old, resulting in reduced quality of life, increased healthcare costs and accident-related death [1–3]. If at-risk individuals can be identified prior to injury, the likelihood of falling can be significantly reduced through intervention such as physical therapy [4,5]. Fall risk generally results from instability arising from neuromuscular deficiencies or external perturbations. Clinical assessments for identifying individuals who would benefit from preventative care are currently limited to questionnaires [6,7] and non-perturbative motor assessments [8–10]. Self-reported information often has low reliability [11,12], and current clinical motor performance tests have low fall-prediction rates, especially for active older adults [13–16]. For widespread use, stability assessments must combine predictive power with minimal experimental and computation time.

Because older adults are more likely to fall while in motion [17], several studies suggest that quantifying dynamic stability may help identify biomechanical deficiencies associated with an increased risk of falling [18–21]. Thus, a number of model-based methods have been developed to assess stability during walking [21]. Among these, variability measures [22] and the maximum Lyapunov exponent [23] ranked highest overall in validity. However, these metrics only characterize a subject's ability to recover from small perturbations. Currently, it is unclear whether the most useful estimates of stability are provided by measuring an individual's ability to recover from small perturbations or by computing the largest possible perturbation that they can withstand without failure of a control strategy [21]. Unfortunately, these methods cannot yet be fully compared because a verified technique for computing the maximum perturbation from which a subject can recover does not exist. Additionally, the best-performing model-based methods are limited to periodic motion.

Stability during non-rhythmic motion is of interest, especially because difficulty with aperiodic tasks such as Sit-to-Stand is strongly correlated with falls in older adults, and because these tasks are necessary for maintaining independence and quality of life [24]. The motion of a person's centre of mass (COM) during Sit-to-Stand can be modelled as an inverted pendulum, which requires an appropriate amount of angular momentum at seatoff to successfully complete the task [25]. Drawing from this idea, metrics of stability for Sit-to-Stand are generally based on an individual's initial COM velocity or acceleration [26–28]. However, a stability metric that considers data only at the onset of movement disregards valuable information about the control strategy used by the individual. In fact, human control strategies for Sit-to-Stand lie on a spectrum ranging from Quasi-Static, in which little momentum is used and the body position is statically stable throughout the motion, to Momentum-Transfer, which is statically unstable and extensively uses momentum to achieve standing [29].

To account for the dynamic differences in Sit-to-Stand motions under distinct control strategies, Shia *et al.* introduced the Stability Basin, computed using individualized pendulum models of Sit-to-Stand with linear-quadratic regulator (LQR) controllers [30]. For this pendulum model, the Stability Basin at a given time is equal to the set of pendulum states that are able to successfully achieve standing under the specified controller. Though this Stability Basin under an LQR controller successfully distinguishes between less and more stable Sit-to-Stand strategies, its ability to accurately identify the set of states that can arrive at standing without failure of a given control strategy is unverified.

To test whether the Stability Basin can accurately estimate an individual's stability for a particular task, perturbations must be introduced in a way that causes the states of the individualized dynamic model to exit the stable region. By stability, we mean the set of body configurations through time from which a subject can successfully stand up without switching from their chosen control strategy to stepping or sitting. An accurate prediction of instability corresponds to an experimentally observed failure of the individual's control strategy.

Here we use a perturbative Sit-to-Stand experiment with 11 subjects, described in §2.1, to validate stability predictions generated by the Stability Basin method. Study participants attempted to complete the Sit-to-Stand motion with their natural, Quasi-Static and Momentum-Transfer control strategies while subjected to forwards or backwards cable pulls, applied to their approximate COM. We use a dynamic model, presented in §2.2, and subject-specific controllers to form individualized Stability Basins, detailed in §2.4. Then, we test whether each subject's Stability Basin can accurately

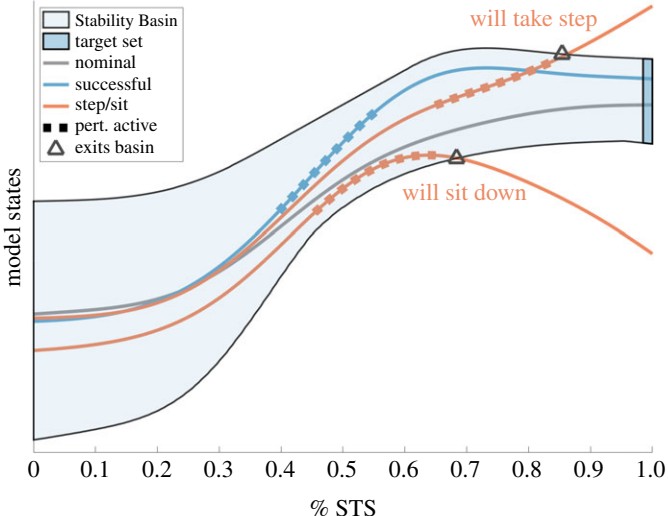

**Figure 1.** An illustrative overview of Stability Basins for Sit-to-Stand. The Stability Basin represents the set of model states through time that will successfully arrive at a target set for a given individual and Sit-to-Stand strategy. Trajectories of the model are illustrated, where the times that perturbations are applied are denoted by the dashed lines. Trajectories that exit the Stability Basin are predicted to lead to stepping or sitting.

predict the failure of a control strategy to recover from the cable pull, described in §2.5. We consider common failure modes, such as a step or sit, to be observed failures of a Sit-to-Stand control strategy [31]. In contrast to prior work that assumed that humans use LQR control during Sit-to-Stand while constructing the Stability Basin, we use a data-driven method to create bounds on the control input for a particular subject and control strategy. Two existing control models and our proposed Input Bounds method are described in §2.3. Across all subjects and control strategies, the proposed method correctly predicts unsuccessful trials over 90% of the time, and correctly identifies successful trials over 95% of the time in a leave-one-out assessment. When compared to three other methods of assessing Sit-to-Stand stability, the proposed method out-performed each alternative. This perturbative validation experiment demonstrates that the Stability Basin method is accurate and reliable, showing promise for integration into current healthcare infrastructure.

## 2. Methods

The goal of this work is to characterize stability during Sit-to-Stand. We hypothesize that Stability Basins can accurately represent stability. The Stability Basin can be understood intuitively as follows. Given a dynamical model of Sit-to-Stand, kinematic observations of a subject performing Sit-to-Stand can be used to form trajectories of the model. The Stability Basin defines a boundary between 'successful' and 'unsuccessful' trajectories of the model, where 'unsuccessful' implies the subject will have to take a step or sit back down. The Stability Basin is introduced pictorially in figure 1, with a formal definition and computational details given in §2.4. An accurate Stability Basin should predict when a subject's Sit-to-Stand motion will be successful or unsuccessful based on model trajectories alone. To test the accuracy of Stability Basins, we performed a perturbative Sit-to-Stand experiment, occasionally inducing subjects to step or sit back down, and then compared Stability Basin predictions to experimentally observed outcomes.

This section describes our framework for computing and testing the accuracy of Stability Basins. First, we collect kinematic observations of a subject performing a perturbative Sit-to-Stand experiment (§2.1). Individualized biomechanical models of Sit-to-Stand are constructed for each subject (§2.2), and three controller models are detailed in §2.3. Our proposed controller, which uses strategy-specific input bounds, reflects the distinct range of inputs required for each control strategy [32]. After computing the Stability Basins (§2.4) for each controller model, we test whether the individual and strategy-specific Stability Basins correctly predict when a subject steps or sits down in response to perturbation, and when they are successful (§2.5). Finally, we perform a comparison to a naive method for estimating stability (§2.4.3).

## 2.1. Perturbative Sit-to-Stand experiment

### 2.1.1. Experimental protocol

Subjects began in a seated position on a stool with their arms crossed. The height of the stool was adjusted so that the subject's thighs were parallel to the ground. Subjects practised standing up from the stool and were asked to find a comfortable foot position, which was then demarcated with a line of tape.

Subjects were instructed to perform three different Sit-to-Stand control strategies: their natural strategy, a Momentum-Transfer strategy and a Quasi-Static strategy as done by Shia *et al.* [30]. Subjects watched a demonstration of each strategy, and then practised each strategy a minimum of 10 times prior to data collection. The following set of treatments were applied to each Sit-to-Stand strategy:

— *Nominal trials*. Subjects stood five times from a comfortable foot position using the specified control strategy.
— *Foot-shift trials*. Subjects varied their foot placement in 0.05-m (2-inch) increments in the anterior–posterior direction from their original position, which were demarcated by taped lines on the ground. Subjects stood up once at each increment, moving their feet backwards until their heels left the ground, and then forward until a strategy shift was observed. We noticed that subjects at extreme anterior foot positions attempted to stand up by lunging forward into a squatting position, and excluded these trials from our dataset because they represented a shift in strategy from the subject's nominal behaviour.
— *Cable pull perturbations*. Subjects returned their feet to their original position. A cable system was attached to the subject's waist and connected to two high-torque motors. These motor driven cables applied impulses to the subject as they rose. The cable pulls were restricted to the anterior–posterior direction, and were applied either forward or backwards with variable timing and force. Specifically, three peak force levels—low, medium and high—were calibrated to each subject. The low force level was designed to rarely induce stepping or sitting during Sit-to-Stand, while the high force level was designed to induce stepping or sitting approximately half of the time. Six trials were taken at each force level for each Sit-to-Stand strategy, with three pulling forwards and three pulling backwards, in random order. For more details, see appendix A.

We collected 948 trials from 11 participants (three female and eight male; ages 18–32; height 1.70 ± 0.12 m; body mass 65.4 ± 10.2 kg), including 163 nominal, 194 foot-shifted and 591 cable pull trials. Each subject gave their informed written consent, and had no physical or balance disorders which could affect their ability to perform Sit-to-Stand.

### 2.1.2. Kinematic observations

A 10-camera PhaseSpace motion capture system collected kinematic observations of 36 markers at 480 Hz. C-Motion's Visual3D biomechanics software was used to fit body segment models to each subject's data [33]. MATLAB was used for all subsequent analyses [34]. A 6th-order Butterworth filter with a cut-off frequency of 2 Hz was used to filter joint position trajectories.

We estimate the motion of the subject's centre of mass (COM) from these kinematic data. First, we use a 3-segment model to track the motion of the subject's shank, thigh and head-arms-torso segments in the sagittal plane. The approximate COM of each segment is computed using anthropometric data [35], and combined to find the trajectory of the total body's COM position throughout Sit-to-Stand. Then, we obtain COM velocity and acceleration trajectories by numerically differentiating the COM position trajectories.

### 2.1.3. Classifying trials as successful/unsuccessful

Two clear instances of a subject's chosen control strategy (natural, Momentum-Transfer or Quasi-Static) failing during Sit-to-Stand are taking a step and sitting back down. We refer to the trials in which these occurred as *steps* and *sits*, and collectively label them *unsuccessful*. At times, we will also refer to these trials as *failures*.

For step trials, the instance of failure is defined as the time when the subject moves the toes of either foot more than 0.0762 m (3 inches) in the anterior–posterior direction from its starting point, as measured by motion capture markers placed on the subject's foot. Although previous studies measured ground reaction forces to identify the onset of steps [31], the distance threshold was used here to use motion

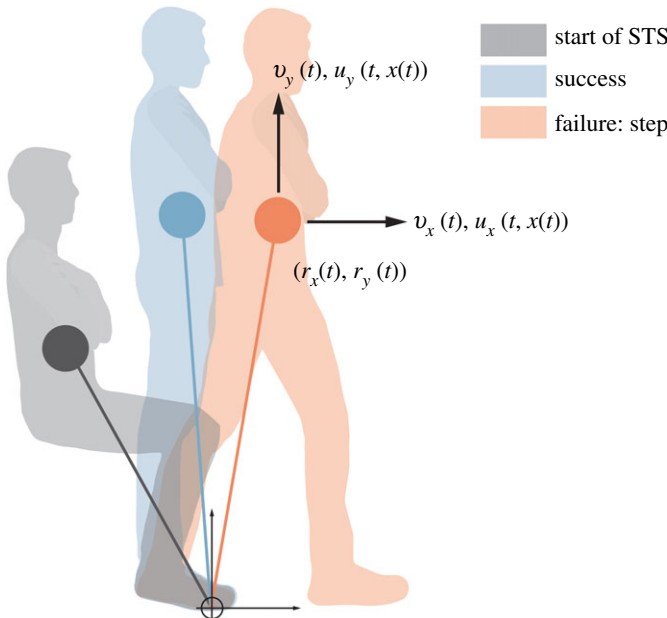

**Figure 2.** Subjects began from a seated position with their arms crossed against their chests. The subject's COM is illustrated by filled circles. Cable-pulls applied to the subject sometimes caused them to step or sit; otherwise, the trial was considered to be successful.

capture data. For sit failures, the failure instance is defined as the time when both horizontal and vertical velocities of the COM become negative, i.e. the COM begins to move back towards the seat. If a subject both stepped and sat down, failure is defined as the earlier of the two.

All foot-shift trials were considered successful. Subjects attempting the extreme anterior and posterior foot positions were either unable to initiate the Sit-to-Stand movement, or violated the parameters of acceptable trials stated in §2.1.1. In either case, these trials were not included in the dataset.

If no step or sit is detected, the trial is considered *successful*. In this work, 'successful trials' refers to nominal trials, foot-shift trials and successful perturbed trials. Note, this scheme classifies trials in which subjects rocked far onto their heels or toes to maintain balance as successful, so long as no step or sit occurred.

## 2.2. Dynamic modelling

To describe our dynamic Sit-to-Stand model and trajectory formation, we adopt the following mathematical notation. Let $A \times B$ denote the Cartesian product of sets A and B. Let $\mathbb{R}^n$ be the $n$-dimensional Euclidean space, and let $\tau \in [0, T] \subset \mathbb{R}$ denote a time drawn from a time interval. Without loss of generality, we assume each Sit-to-Stand trial starts at time $\tau = 0$.

### 2.2.1. Telescoping Inverted Pendulum Model

We use a Telescoping Inverted Pendulum Model (TIPM, figure 2) to model each subject's COM motion in the sagittal plane during each Sit-to-Stand trial [36]. This model is capable of describing the sizable displacements of a subject's COM in the horizontal and vertical directions that occur during Sit-to-Stand. The TIPM consists of a point mass of mass $m \in \mathbb{R}$ representing the subject's COM. Individualized TIPMs are constructed for each subject by setting $m$ as the subject's mass.

We define the origin as the initial position of the ball of the subject's foot. Specifically, the origin is defined as the mean initial position of motion capture markers attached to the metatarsophalangeal joints of the subject's left and right feet, as estimated by Visual 3D. We let the subscript $(\cdot)_x$ denote a quantity in the anterior–posterior direction, and the subscript $(\cdot)_y$ denote a quantity in the vertical direction. At each time $\tau \in [0, T]$, we denote the following quantities of the TIPM as:

— positions: $\tilde{r}_x(\tau), \ \tilde{r}_y(\tau) \in \mathbb{R}$
— velocities: $\tilde{v}_x(\tau), \ \tilde{v}_y(\tau) \in \mathbb{R}$
— accelerations: $\tilde{a}_x(\tau), \ \tilde{a}_y(\tau) \in \mathbb{R}$

— inputs: $\tilde{u}_x(\tau),\ \tilde{u}_y(\tau) \in \mathbb{R}$
— cable-pull forces: $\tilde{d}_x(\tau),\ \tilde{d}_y(\tau) \in \mathbb{R}$.

Let $\tilde{x}:[0, T] \to X \subset \mathbb{R}^4$ be a state trajectory, so that $\tilde{x}(\tau) \in X$ is the model's state at time $\tau \in [0, T]$:

$$\tilde{x}(\tau) = \begin{bmatrix} \tilde{r}_x(\tau) \\ \tilde{v}_x(\tau) \\ \tilde{r}_y(\tau) \\ \tilde{v}_y(\tau) \end{bmatrix}. \tag{2.1}$$

Let $\tilde{u}:[0, T] \times X \to U \subset \mathbb{R}^2$ be an input trajectory, so that $\tilde{u}(\tau, \tilde{x}(\tau))$ is the input at that time and state:

$$\tilde{u}(\tau, \tilde{x}(\tau)) = \begin{bmatrix} \tilde{u}_x(\tau, \tilde{x}(\tau)) \\ \tilde{u}_y(\tau, \tilde{x}(\tau)) \end{bmatrix}. \tag{2.2}$$

The time derivative of $\tilde{x}(\tau)$ at time $\tau$, denoted $\dot{\tilde{x}}(\tau)$, can be written as:

$$\dot{\tilde{x}}(\tau) = \begin{bmatrix} \tilde{v}_x(\tau) \\ \frac{1}{m}(\tilde{u}_x(\tau, \tilde{x}(\tau)) + \tilde{d}_x(\tau)) \\ \tilde{v}_y(\tau) \\ \frac{1}{m}(\tilde{u}_y(\tau, \tilde{x}(\tau)) + \tilde{d}_y(\tau)) - g \end{bmatrix}, \tag{2.3}$$

where $g$ is the gravitational acceleration $9.81\ \mathrm{m\,s^{-2}}$.

### 2.2.2. Nondimensionalizing time

The time that it takes to complete Sit-to-Stand varies from trial to trial. Details on how the start and end of each Sit-to-Stand trial are chosen are provided in appendix B. To make accurate comparisons across trials, we normalize each TIPM state trajectory $\tilde{x}$ by the trial's length, so that each trajectory occurs over the interval $[0, 1]$, which we refer to as 0 to 100% STS. We introduce a unitless time variable $t$, related to $\tau$ by $t = \tau/T$. This dimensionless variable allows us to introduce a normalized state trajectory $x : [0, 1] \to X$, where

$$x(t) = \begin{bmatrix} \tilde{r}_x(T \cdot t) \\ T \cdot \tilde{v}_x(T \cdot t) \\ \tilde{r}_y(T \cdot t) \\ T \cdot \tilde{v}_y(T \cdot t) \end{bmatrix}. \tag{2.4}$$

We scale accelerations and forces by a factor of $T^2$ following similar logic.

For the rest of this paper, we only consider normalized trajectories, unless explicitly stated. Let $r_x$, $r_y$, $v_x$ and $v_y$ refer to normalized position and velocity trajectories, and let $a_x$, $a_y$, $u_x$, $u_y$, $d_x$ and $d_y$ refer to normalized acceleration and force trajectories.

## 2.3. Controller models

The Stability Basin computation, detailed in §2.4, requires a model for the TIPM controller. In this subsection, we describe and compare two existing controller models as well as our proposed controller model. All three controller models are used to compute Stability Basins that are compared during the validation phase described in §2.5.3. Before we describe these controller models in mathematical detail, we briefly describe their qualitative properties. The controllers are compared and contrasted in figure 3.

The controller models estimate the control input $u(t, x(t))$ at a given time $t$ and state $x(t)$. In many previous studies, researchers have modelled human controllers as an open-loop 'feed-forward' component plus a closed-loop linear 'feedback' component [30,37,38], thus finding a single controller that best models the data. Following this paradigm, we develop an LQR controller model and a traditional feed-forward plus feedback controller model (FF+FB). The main differences between these controllers are the specifics of their construction, detailed in §2.3.2 and §2.3.3. The LQR controller model is constructed by solving an optimal control problem to provide feedback about an average nominal trajectory, while the FF+FB controller model is constructed via linear regression. Both of these controller models return a single input at a given time and state.

Because the form of the controller employed by humans is unknown, it is difficult to assert with confidence that a single controller adequately captures human behaviour. We propose that a

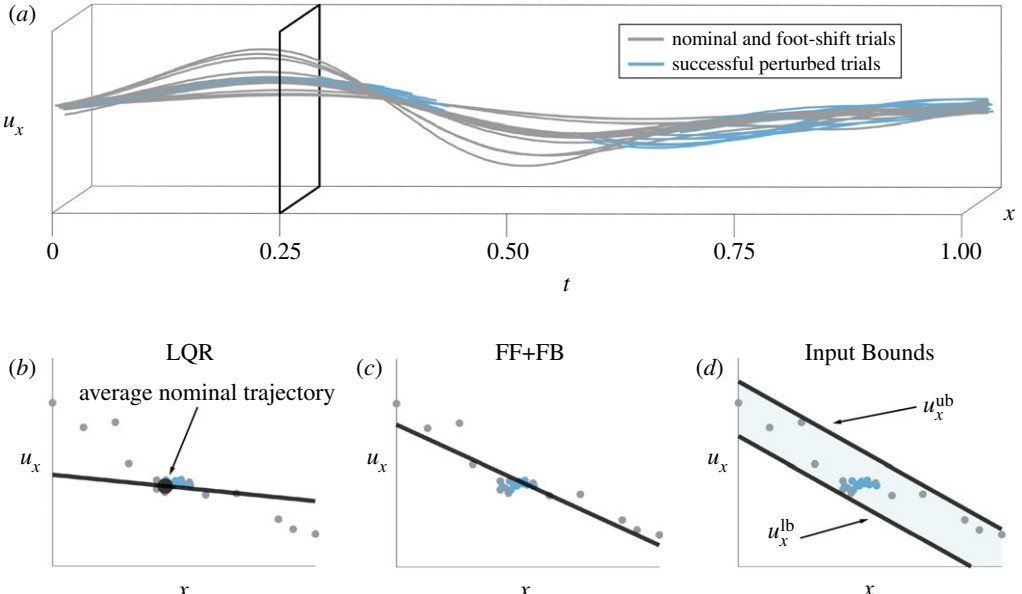

**Figure 3.** These figures show how we construct the controller models from data. The process is shown for $u_x$, and is identical for $u_y$. In (a), $u_x$ evolves as a function of time $t$ and state $x$, which has been depicted as a single dimension for this illustration. The grey lines represent the input trajectories of nominal and foot-shift trials, while the blue lines represent the input trajectories of perturbed successful trials. Note that we have not plotted the portions of the successful perturbed trajectories when the perturbations were active, because these segments are not used to train the controller models. An arbitrary time $t = 0.25$ is illustrated by the black outline in (a). The observed inputs from nominal, foot-shift and perturbed successful trials at that time are plotted as points in (b–d). (b) shows that the LQR controller uses linear feedback about a nominal trajectory. Because the LQR controller is generated by minimizing a cost function specified by the matrices $Q$ and $R$, its feedback matrix does not necessarily coincide with a line of best fit. (c) illustrates that the FF+FB controller is given by a line of best fit. (d) shows how the upper and lower bounds $u_x^{lb}(t, x(t))$ and $u_x^{ub}(t, x(t))$ are parallel linear functions of state that encompass all of the successful inputs.

more robust controller model may return a set of possible inputs rather than a single input at a given time and state. One way to achieve this is to bound the possible inputs at each time and state, and assume that the human could use any input within these bounds. We refer to this controller model as the *Input Bounds* controller.

The Input Bounds represent the range of time-varying inputs that are expected under a given Sit-to-Stand strategy. For example, consider a subject's natural Sit-to-Stand strategy. At the beginning of the motion, we expect the subject to apply a large positive horizontal input to propel themselves forwards, while applying little positive vertical input. Halfway through the motion, the subject may apply a negative horizontal input to slow their forward velocity, as well as a large positive vertical input to rise towards standing. Therefore, the expected inputs applied by the subject are time-dependent. Moreover, because there is some variability in the way the subject performs Sit-to-Stand from trial to trial, we expect the inputs to fall within some range of values. The Input Bounds controller captures this variability by developing time-dependent upper and lower bounds on the allowable inputs.

Biomechanical constraints, such as ground reaction force limits or joint torque limits, constrain the maximum inputs that a subject can apply during Sit-to-Stand under any strategy. We want to be clear that the Input Bounds *do not* represent the range of all inputs that are biomechanically feasible, but rather the subset of inputs that represent the range expected under a given strategy. For example, consider a scenario in which a person begins standing up using their natural Sit-to-Stand strategy, but then stops halfway and remains in a crouched position. Certainly, the person can achieve this performance without violating biomechanical constraints. However, the person will have deviated from their natural Sit-to-Stand strategy, in which they normally stand all the way up. Accordingly, the input that they applied would have exited the Input Bounds we have developed for their natural strategy at the time that they decided to remain in a crouched position. For this reason, we develop the Input Bounds from observations of Sit-to-Stand rather than from biomechanical constraints.

### 2.3.1. Generating training data

Modelling human motion requires the LQR, FF+FB and Input Bounds controller models to be estimated from observed data. In this work, we use kinematic observations of successful trials (i.e. nominal, foot-shift and successful perturbed trials) to generate a dataset on which to train each control model. Note that the dataset on which each controller model is trained is identical. We fit the LQR, FF+FB and Input Bounds controllers separately for each subject's natural, Momentum-Transfer and Quasi-Static control strategies.

Given a strategy, we first compute the inputs for each successful trial within that strategy via inverse dynamics using (2.3) [39, Chapter 5]. Let $x_i(t)$ and $u_i(t)$ denote the state and input at time $t$ for the $i$th successful trial computed via inverse dynamics, and let $S$ denote the set of successful trials within a given strategy. Given a time $t$, we will frequently use the notation

$$\sum_{i \in S} (u(t, x_i(t)) - u_i(t))^2 \tag{2.5}$$

which means 'the squared error of the estimated control input $u(t, x_i(t))$ from each observed control input $u_i(t)$, summed over all successful trials within a given strategy'.

Because many successful trials within a Sit-to-Stand strategy involved cable-pull perturbations, we make the following remark:

**Remark 2.1.** When computing $u_i(t)$ via inverse dynamics for successful perturbed trials, we discard the portion of the trajectory during which the perturbation was active. This is possible because the perturbations were time-synchronized with the motion capture data.

### 2.3.2. LQR controller

Shia *et al.* proposed to model Sit-to-Stand using an LQR controller about a nominal Sit-to-Stand trajectory [30]. To build this controller, we first form a single average nominal trajectory $\bar{x}$ for each of a subject's Sit-to-Stand control strategies by taking the mean of the five nominal Sit-to-Stand trajectories. We generate an open loop controller, $u_{ol} : [0, 1] \to U \subset \mathbb{R}^2$, for each average nominal trajectory via inverse dynamics. Then, we use LQR to design a linear feedback controller about the average nominal trajectory. We specify the quadratic cost function for LQR by a state weighting matrix $Q \in \mathbb{R}^{4 \times 4}$ and input weighting matrix $R \in \mathbb{R}^{2 \times 2}$, which are used to generate the feedback matrix $K(t) \in \mathbb{R}^{2 \times 4}$. [40, Chapter 16] At a given time and state, the input from the LQR controller can then be written as:

$$u(t, x(t)) = u_{ol}(t) - K(t)(x(t) - \bar{x}(t)). \tag{2.6}$$

We choose the $Q$ and $R$ that produces a controller that most closely fits the observed data. To do so, we wrap the choice of $Q$ and $R$ inside an optimization procedure that seeks to minimize the error between the LQR controller model and the observed control inputs over the time horizon [0, 1]:

$$\min_{Q,R} \quad \sum_{i \in S} \int_0^1 (u(t, x_i(t)) - u_i(t))^2 \, \mathrm{d}t. \tag{2.7}$$

As in (2.5), $S$ represents the set of observed successful trials within a Sit-to-Stand strategy, and $x_i(t)$ and $u_i(t)$ correspond to the observed states and input of the $i$th trial at time $t$. The integral is approximated using 200 time steps. We specify the space over which $Q$ is optimized:

$$Q \in \{M \in \mathbb{R}^{4 \times 4} | M_{i,j} = 0 \ \forall i \neq j, \ M_{1,1} = 1, \ M_{2,2} \in [0.1, 100], \ M_{3,3} \in [0.1, 100], \ M_{4,4} \in [0.1, 100]\} \tag{2.8}$$

and the space over which $R$ is optimized:

$$R \in \{M \in \mathbb{R}^{2 \times 2} | M_{i,j} = 0 \ \forall i \neq j, \ M_{1,1} \in [1e-5, 1e-2], \ M_{2,2} \in [1e-6, 1e-1]\}. \tag{2.9}$$

$Q$ and $R$ are constant diagonal matrices, where the spaces over which each is optimized were chosen to restrict the relative costs of states versus inputs, the relative costs between states, and the relative costs between inputs. In particular, no state can cost more than 100 times any other state, no input can cost more than 10 times another input, and the relative scale between states and inputs was chosen based on the expected magnitudes of states and inputs.

### 2.3.3. FF + FB controller

We also test the efficacy of a traditional feed-forward plus feedback controller. Let the input $u(t, x(t))$ at some time $t$ and state $x(t)$ be described by the state feedback matrix $K(t) \in \mathbb{R}^{2 \times 4}$ and a feed-forward component ff($t$):

$$u(t, x(t)) = \text{ff}(t) - K(t)x(t). \tag{2.10}$$

To generate $K(t)$ and ff($t$) from data, we use linear least squares to solve the following program:

$$\min_{\text{ff}(t), K(t)} \quad \sum_{i \in S} (u(t, x_i(t)) - u_i(t))^2. \tag{2.11}$$

As in (2.5), $S$ represents the set of observed successful trials within a Sit-to-Stand strategy, and $x_i(t)$ and $u_i(t)$ correspond to the observed states and input of the $i$th trial at time $t$.

### 2.3.4. Input Bounds controller

The proposed Input Bounds controller differs from both the LQR and FF+FB controllers by returning a set of inputs at a given time and state, rather than a single point. To define the Input Bounds controller, we develop bounds on the TIPM control input. We denote a lower bound on the control input as $u^{\text{lb}}$, and an upper bound on the control input as $u^{\text{ub}}$. Note that

$$u^{\text{lb}} \leq u \leq u^{\text{ub}}, \tag{2.12}$$

where the inequality is taken element-wise.

We present this approach more formally in the following definition:

**Definition 2.2.** For the Input Bounds controller, we model the inputs $u$ to the TIPM as a bounded component plus a linear feedback component, where the bounded component is drawn from a bounded set. Specifically, given time $t$ and state $x(t)$, we define the bounds in (2.12) as

$$u^{\text{lb}}(t, x(t)) = b^{\text{lb}}(t) - K(t)x(t) \tag{2.13}$$

and

$$u^{\text{ub}}(t, x(t)) = b^{\text{ub}}(t) - K(t)x(t), \tag{2.14}$$

where $b^{\text{lb}}(t)$ and $b^{\text{ub}}(t) \in \mathbb{R}^2$ are lower and upper bounds on the bounded component, and $K(t) \in \mathbb{R}^{2 \times 4}$ is a matrix of linear feedback gains.

We generate the parameters of the Input Bounds controller $b^{\text{lb}}$, $b^{\text{ub}}$ and $K$ from data by solving a constrained linear least-squares problem:

$$\min_{b^{\text{lb}}(t), b^{\text{ub}}(t), K(t)} \quad \sum_{i \in S} (u^{\text{lb}}(t, x_i(t)) - u_i(t))^2 + (u^{\text{ub}}(t, x_i(t)) - u_i(t))^2$$

$$\text{s.t.} \quad u^{\text{lb}}(t, x_i(t)) \leq u_i(t) \qquad \forall i \in S,$$

$$u^{\text{ub}}(t, x_i(t)) \geq u_i(t) \qquad \forall i \in S$$

where the inequalities are understood element-wise. This is a quadratic program, which can be solved rapidly and efficiently.

In summary, at each instance in time, we compute the range of control inputs of a certain form that can be applied while explaining the data we observed, thereby modelling a range of inputs a human may use to recover from perturbation. Our method of Stability Basin computation, detailed in §2.4.2, is flexible to this type of controller specification, and in particular, we verify that this approach is not too conservative in §3. In effect, the Input Bounds controller model allows us to compute Stability Basins with a set of hypothetical controllers, instead of just one.

## 2.4. Computing Stability Basins

Three elements are required to form the Stability Basin for each of an individual's Sit-to-Stand control strategies:

1. The TIPM dynamics (2.3).
2. A controller model for the TIPM (e.g. (2.6), (2.10), (2.12)).

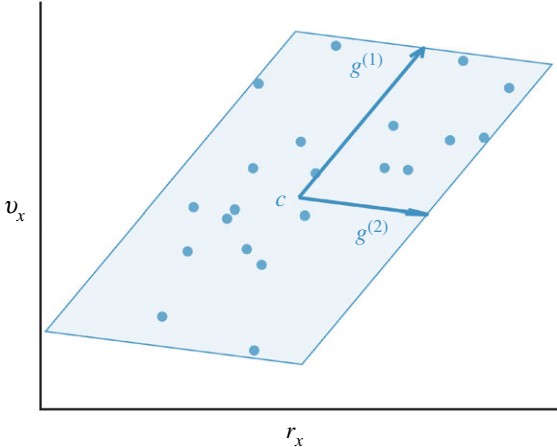

**Figure 4.** This figure illustrates how $X_T$ (the target set) is formed from data for a given subject and strategy. The states ($r_x$ and $v_x$) at 100% STS of each observed successful trial within a strategy are illustrated as blue dots. An example $X_T$ represented by the shaded blue zonotope contains all of the states observed to lead to standing. The zonotope is parametrized by its centre $c$ and generator vectors $g^{(1)}$ and $g^{(2)}$. This example depicts a two-dimensional $X_T$, but in reality they are four-dimensional and encompass both horizontal and vertical components of the state space.

3. A *target set*, $X_T$, that encapsulates all model states observed to lead to successful standing.

Given these elements, we define the Stability Basin as follows:

**Definition 2.3.** The Stability Basin $\subset [0, 1] \times X$ is a subset of times (0–100% STS) and model states from which a trajectory of the model obeying the dynamics (2.3) and controller model will arrive at the target set $X_T$.

In the rest of this subsection, we explain how we compute the target set $X_T$ from data. Then, we use this object in conjunction with a controller from §2.3 to generate the Stability Basins.

### 2.4.1. Generating the target set

The goal of the Sit-to-Stand motion is to arrive at a standing configuration. We define the target set $X_T$ as a set that encompasses the states at time $t = 1$ observed to lead to successful standing. Note, the target set $X_T$ does not represent quiet standing itself, which we consider a distinct task.

To define $X_T$, we must first introduce a geometric object called a *zonotope*. A zonotope $Z$ is a polytope in $\mathbb{R}^n$ that is closed under linear maps and Minkowski sums [41], and is parametrized by its centre $c \in \mathbb{R}^n$ and generators $(g^{(1)}, \ldots g^{(p)})$ shown in figure 4. We write the generators as columns of the generator matrix $G \in \mathbb{R}^{n \times p}$. A zonotope describes the set of points that can be written as the centre $c$ plus a linear combination of the generator matrix columns, where each element of the coefficient vector $\beta \in \mathbb{R}^p$ must be between −1 and 1:

$$Z = \{y \in \mathbb{R}^n \mid y = c + G\beta, \quad -\mathbf{1} \leq \beta \leq \mathbf{1}\}, \tag{2.15}$$

where the inequalities are applied element-wise and $\mathbf{1}$ is a vector of ones of the appropriate size.

We define $X_T \subset X$ as a zonotope that encompasses the final states (i.e. the states at $t = 1$) of all successful trials observed for a given Sit-to-Stand strategy. Specifically, $X_T$ is a zonotope with four generators computed from the final states as proposed by Stursberg [42, §3], where each generator is expanded by 5% to avoid observed states lying on the edge of the set. The target set generation process is depicted in figure 4.

### 2.4.2. Computing Stability Basins via reachability analysis

The reader may notice that the Stability Basin (definition 2.3) is defined as a set of states through time that satisfy some dynamics specified by an ordinary differential equation (ODE). Generally, numerical integration can be used to find solutions of an ODE starting from a single point. As we are interested in finding all possible solutions of the ODE, subject to constraints, we use a method called reachability analysis to compute the Stability Basins. Rather than flowing a *single point* through a

vector field, reachability analysis flows a *set* through a vector field. The set of all states that can be attained by the system, starting from some initial set, is referred to as the *forwards reachable set*. Conversely, all states that can end in some final set are referred to as the *backwards reachable set*.

To compute the Stability Basins, we use an open-source reachability toolbox called CORA, which efficiently handles difficulties associated with representing set dynamics [43,44]. CORA represents the Stability Basin as a zonotope at each of a finite collection of *time steps*, which are subintervals of the interval [0, 1] of length $\Delta t$. With a slight abuse of notation, we let $t$ act as an index, so that $Z^{(t)}$ is the zonotope representing the Stability Basin over the time step that contains $t$. Briefly, CORA works by linearizing the dynamics at each time step about the centre of $Z^{(t)}$, and obtaining $Z^{(t+\Delta t)}$ by multiplying $Z^{(t)}$ by an overapproximation of the matrix exponential over that time step. It then expands $Z^{(t+\Delta t)}$ to account for the effects of inputs and linearization error. Each Stability Basin was formed using a time step $\Delta t$ of 0.005 (i.e. 0.5%STS), so that 200 zonotopes represent the Stability Basin over the interval [0, 1]. Although CORA chooses the exact number of generators to use for each zonotope, we set the maximum number of generators as 800.

Generally, CORA expects the inputs to the system to be functions of time, allowing the feed-forward inputs and gain matrices for the LQR and FF+FB controllers to be specified at each time step. However, CORA also allows the input to the system at each time step to be drawn from a set, allowing us to use the Input Bounds controller model and let the control input be defined as in (2.13) and (2.14). CORA in this case assumes the input at a given time and state can take any value within the Input Bounds to create the Stability Basin. Because the Stability Basin is the set of states through time that arrive at the final target set $X_T$, it can be computed as a backwards reachable set. In practice, this means we treat $X_T$ as an initial set, and use CORA to flow the set *backwards* in time under the negative of the dynamics (2.3) and the controller model.

Stability Basins were generated for each controller described in §2.3 on a laptop computer with a 2.7 GHz Intel Core i7 processor. Stability Basins computed using the Input Bounds controller take 0.76 ± 0.014 seconds to compute.

### 2.4.3. Naive method for computing Stability Basins

Stability can also be estimated from observed perturbed trials by simply drawing a volume around the state trajectories of the observed successful trials. This method does not use reachability or a controller model, and relies on state trajectories alone. Here, we apply this method by generating a four-dimensional zonotope at each time step that encloses all of the observed successful states. This is the same procedure that we employ to generate the target set $X_T$ (described in §2.4.2 and illustrated in figure 4) applied at each instance in time over the course of the motion.

## 2.5. Evaluating Stability Basin accuracy

We hypothesize that the Stability Basins can predict when a subject must depart from their control strategy in response to perturbation to avoid falling. Previously in §2.1.3, we defined the failure of a control strategy as stepping or sitting back down and explained how we determine the onset of stepping or sitting. We now use the Stability Basins to predict whether or not a strategy failure will occur by checking if a Sit-to-Stand trajectory exits the Stability Basin at any point. Finally, we detail the evaluation procedure we used to test the accuracy of the Stability Basins' predictions.

### 2.5.1. Defining onset of failure

To test the predictive power of the Stability Basins, we first identify the onset of failure (as previously described in §2.1.3) during a Sit-to-Stand movement. We denote $t_f \in [0, 1]$ as the time of failure onset, and will test whether the Stability Basins predict failure prior to that time. Because we are using a procedure based on an average nominal trial for aligning and segmenting trials (detailed in appendix B), it is possible for the onset of stepping or sitting to occur after the trial end time $t = 1$. The end times chosen by the segmentation procedure do not necessarily imply that the subject has reached standing, but are just a means to achieve a consistent segmentation of the data. We consider a trial with a step or sit occurring later than the trial's end to be unsuccessful, and set $t_f = 1$ in this case.

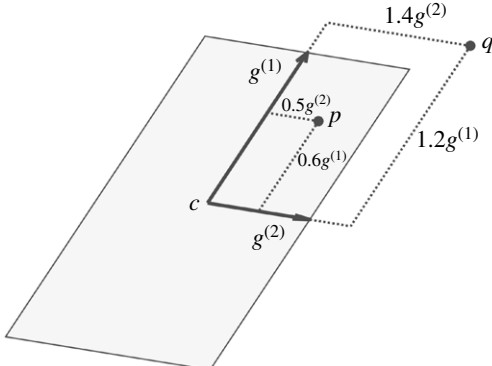

**Figure 5.** This figure illustrates how to check if a point is inside or outside of a zonotope. An example zonotope (shaded grey) is parametrized by its centre $c$ and generator vectors $g^{(1)}$ and $g^{(2)}$. A test point $p$ is contained within the zonotope because the maximum absolute value of the coefficients on $g^{(1)}$ and $g^{(2)}$ is less than or equal to 1. Another test point $q$ lies outside of the zonotope because the maximum absolute value of the coefficients on $g^{(1)}$ and $g^{(2)}$ is greater than 1.

### 2.5.2. Checking trajectories

Given an observed trajectory $x_i$ from a Sit-to-Stand trial, we use a linear program to determine whether that trajectory lies inside the Stability Basin at each time step. Recall that each zonotope $Z^{(t)}$ representing the Stability Basin is parametrized by its centre $c^{(t)} \in \mathbb{R}^4$ and generator matrix $G^{(t)} \in \mathbb{R}^{4 \times p}$. At each time $t$, we represent the state $x_i(t)$ as a linear combination of the generator matrix's columns, and use a linear program to find the representation that has the smallest maximum coefficient $\beta_{\max}^{(t)} \in \mathbb{R}$:

$$\min_{\beta_{\max}^{(t)} \in \mathbb{R}, \beta^{(t)} \in \mathbb{R}^p} \quad \beta_{\max}^{(t)}$$
$$\text{s.t.} \quad G^{(t)} \beta^{(t)} = x_i(t) - c^{(t)}, \tag{2.16}$$
$$|\beta^{(t)}| \leq \beta_{\max}^{(t)}$$

where the absolute value and inequalities are applied element-wise. If $\beta_{\max}^{(t)}$ is less than or equal to 1, the point $x_i(t)$ is inside the Stability Basin at that time step, depicted in figure 5.

### 2.5.3. Evaluation procedure

The Stability Basins are used to predict the success or failure of every trial within each Sit-to-Stand strategy (natural, Momentum-Transfer and Quasi-Static). The accuracy of each Stability Basin is then tested by comparing its predictions to the experimentally observed outcomes of each Sit-to-Stand trial.

Given the state trajectory $x_i$ observed for a Sit-to-Stand trial, the linear programs described in (2.16) are used to predict success/failure. If a state trajectory $x_i$ remains inside the Stability Basin at each time step, the trial is predicted to be successful. If $x_i$ exits the Stability Basin at any time step, the trial is predicted to fail. To ensure that the Stability Basins' predictions are made fairly, we make the following restrictions:

1. For cable-pull trials, the prediction uses only the portion of the Stability Basin after the onset of perturbation.
2. For trials in which failure was observed, the prediction uses only the portion of the Stability Basin before failure occurred (i.e. $t \leq t_f$ as in §2.5.1).

We evaluate the Stability Basins formed using the LQR, FF+FB and Input Bounds controllers, as well as the naive method in §2.4.3. We perform separate types of evaluation for successful trials versus unsuccessful trials. For successful trials, we employ a *leave-one-out* procedure to avoid training and testing on the same data. After forming the target set, we leave one successful trial out of the controller (LQR, FF+FB, Input Bounds) training data in §2.3.1. For the naive method, one successful trial is left out of the zonotope generation at each time step. We then construct a Stability Basin using each controller/method to test the trial that was left out. We repeat this procedure until all successful trials have been tested.

For unsuccessful trials, we compute a single Stability Basin for each controller model/method, formed using all observed successful trials. We use this single Stability Basin to predict the outcome of each unsuccessful trial within a given strategy. An outline of the validation procedure is provided in figure 6.

for each subject, strategy (natural, Momentum Transfer, Quasi-Static), and controller/method (Input Bounds, FF+FB, LQR, naive):

(*a*) nominal, foot-shift, and successful perturbed trials

Step 1. Form standing set with all nominal, foot-shift, and successful perturbed trials (figure 4).

Step 2. Train controller (or naive method) with all nominal, foot-shift, and successful perturbed trials except one (figure 3).

Step 3. Form Stability Basin with standing set and controller, if applicable.

Step 4. Check that the left-out trial remains in the basin.

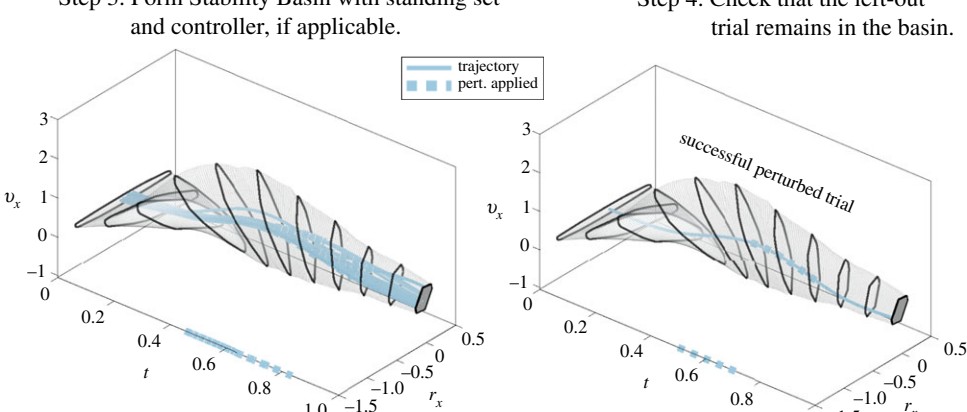

(*b*) unsuccessful trials

Step 1. Form standing set with all nominal, foot-shift, and successful perturbed trials (figure 4).

Step 2. Train controller (or naive method) with all nominal, foot-shift, and successful perturbed trials (figure 3).

Step 3. Form Stability Basin with standing set and controller, if applicable.

Step 4. Check that the unsuccessful trial exits the basin before the step/sit occurs.

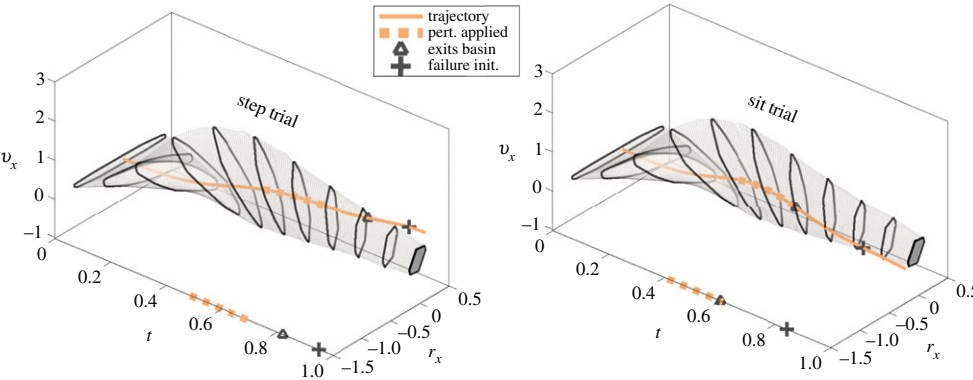

**Figure 6.** An outline of the Stability Basin validation with examples provided by the computed Stability Basin for subject ID 1's natural strategy under the Input Bounds controller. The horizontal projection of the Stability Basin is represented as the region encapsulated by the light grey borders. The projection of the target set $X_T$ is shown as the dark grey region on the right side of each plot. The state trajectories of all of subject ID 1's successful natural strategy trials *except one* are used to construct the Stability Basin on which the left-out trial is tested, as shown in (*a*). Then all of subject ID 1's successful natural strategy trials are used to construct the Stability Basin on which the unsuccessful trials are tested, as shown in (*b*). State trajectories of a step and a sit exit the basin before the onset of failure. As detailed in §2.1.3, we define step initiation as the time when the toes of either foot move more than 0.0762 m (3 inches) in the anterior–posterior direction, and sit initiation as the time at which both $v_x$ and $v_y$ become negative.

To be clear, we do not follow a leave-one-out procedure when generating the target set $X_T$ from data. As displayed in figure 4, the target set generation is similar to taking the convex hull of a set of four-dimensional points. If we follow a leave-one-out procedure when generating the target set, then the points on the boundary of the original set will likely be on the outside of the new set when they

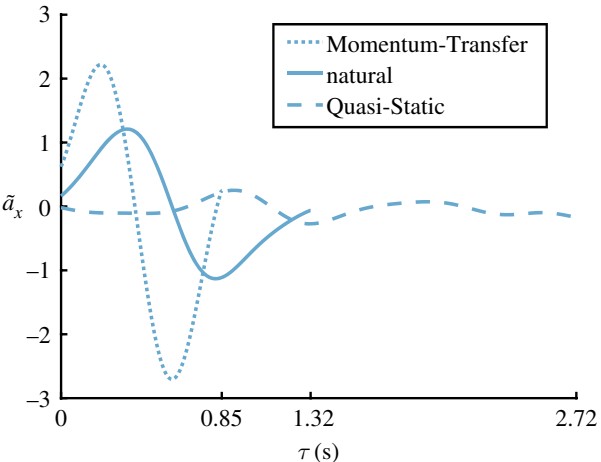

**Figure 7.** This figure shows example nominal unnormalized horizontal acceleration trajectories for each Sit-to-Stand strategy for subject ID 8.

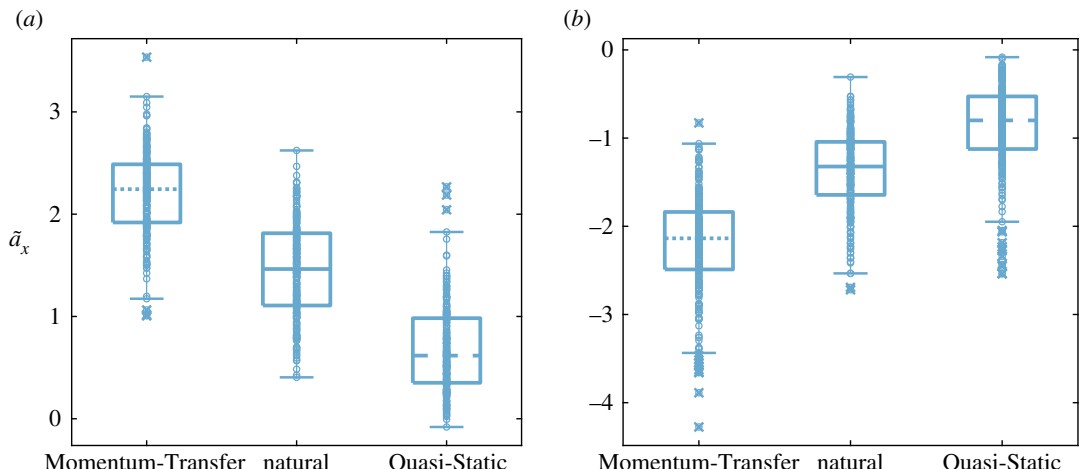

**Figure 8.** (*a*) maximum and (*b*) minimum horizontal accelerations are significantly different between Sit-to-Stand strategies, with $p < 0.001$ when comparing the maximums or minimums of any two strategies under a two-sample $t$-test. Only successful trials are used in this analysis. For each strategy, each small circle represents data from a single Sit-to-Stand trial. The central mark within the box plot indicates the median horizontal acceleration. The bottom and top edges of the box indicate the the 25th and 75th percentiles, respectively. The whiskers extend to the most extreme horizontal accelerations not considered outliers, and the outliers are plotted individually as small circles with an 'x' through them.

are left out. This would mean that many successful trials whose final states lie on the edge of the original target set necessarily exit the Stability Basins, and therefore are automatically misclassified when following this leave-one-out procedure. For these reasons, this procedure would serve to validate the robustness of the target set generation, rather than the Stability Basins themselves. However, collecting a larger number of samples should decrease the effect of missing data on the target set generation.

## 3. Results

The subjects performed Sit-to-Stand trajectories with similar maximum and minimum accelerations for each control strategy as the (different) subjects examined by Shia [30]. Even though subjects were allowed to determine the speed with which each Sit-to-Stand was performed, each subject's natural, Momentum-Transfer and Quasi-Static Sit-to-Stand strategies showed distinct kinematic characteristics, depicted in figure 7. In particular, the maximum and minimum horizontal accelerations of the COM for the Momentum-Transfer ([−2.1, 2.2] m s$^{-2}$) and Quasi-Static ([−0.8, 0.6] m s$^{-2}$) strategies were significantly different from each other as well as from the natural strategy ([−1.3, 1.5] m s$^{-2}$), depicted in figure 8, with

**Table 1.** Statistics collected from the perturbative Sit-to-Stand experiment are aggregated across subjects and summarized below. Note that the onset of failure (Step %STS and Sit %STS) often occurred after the defined trial end ($t = 1$), which is discussed in §2.5.1. Electronic supplementary material that reports these statistics for individual subjects is available online.

| Sit-to-Stand strategy | total | cable pull | steps | sits | trial time (s) | Pert. Onset (%STS) | step (%STS) | sit (%STS) |
|---|---|---|---|---|---|---|---|---|
| natural | 326 | 197 | 42 | 29 | 1.25 ± 0.23 | 0.50 ± 0.10 | 1.07 ± 0.21 | 0.96 ± 0.15 |
| Momentum-Transfer | 322 | 198 | 40 | 19 | 1.13 ± 0.21 | 0.47 ± 0.11 | 1.14 ± 0.33 | 0.99 ± 0.15 |
| Quasi-Static | 300 | 196 | 39 | 29 | 2.23 ± 0.89 | 0.51 ± 0.17 | 0.95 ± 0.21 | 0.85 ± 0.20 |

**Table 2.** The accuracy of the Stability Basins generated using the Input Bounds controller model are reported below. This table reports a tally of the correct predictions for each Sit-to-Stand strategy and trial type across subjects.

| Sit-to-Stand strategy | successful trials | step trials | sit trials |
|---|---|---|---|
| natural | 248/255—97.25% | 41/42—97.62% | 27/29—93.10% |
| Momentum-Transfer | 257/263—97.72% | 33/40—82.50% | 17/19—84.75% |
| Quasi-Static | 213/232—91.81% | 36/39—92.31% | 27/29—93.10% |
| combined | 718/750—95.73% | 110/121—90.91% | 71/77—92.21% |

$p$-values <0.001 when comparing the successful trials within any two strategies. While the duration of natural and Momentum-Transfer trials were similar (average times of 1.25 s and 1.13 s), Quasi-Static trials lasted longer (average 2.23 s), due in part to the low observed COM accelerations. The Stability Basin shape and volume are highly dependent on the kinematic and temporal characteristics corresponding to the strategy selected, as in previous work [30].

When using the Momentum-Transfer strategy during the foot-shift trials, subjects were able to stand with their feet further forward (max. 0.199 ± 0.0699 m) than when using Quasi-Static (max. 0.0417 ± 0.0307 m).

Cable pull perturbations during Sit-to-Stand frequently induced a step or sit to avoid falling. Out of 591 cable pull trials, we observed failure in 198, or 33.5%. Specifically, we observed 42 steps and 29 sits for the natural strategy, 40 steps and 19 sits for Momentum-Transfer and 39 steps and 29 sits for Quasi-Static. These statistics are presented in table 1. Fewer sits were observed for the Momentum-Transfer strategy than in both the natural and Quasi-Static strategies, likely due to the larger forward momentum at seatoff for the Momentum-Transfer strategy.

To assess the accuracy of each Stability Basin, we compared the predictions to the experimental observation for each trial according to the procedure described in §2.5.3. The aggregate prediction rates of the Stability Basins formed using the Input Bounds controller for each Sit-to-Stand strategy across subjects are given in table 2. Across Sit-to-Stand strategies, nominal, foot-shift and successful perturbed trials (figure 6a) remain inside the basin 95.73% of the time. Of the failure trials, 90.91% of steps are predicted prior to onset, while 92.21% of sits are predicted prior to onset (figure 6b). There was no consistent effect of perturbation force on the predictive power of the method, detailed in table 3.

Overall, the Input Bounds Stability Basins' failure predictions are over 90% accurate; only 17 out of 198 unsuccessful trials are incorrectly predicted to be successful. To understand these false successes, we measured the maximum Euclidean distance of each unsuccessful trajectory to its nearest successful trajectory. The false successes are 34.24% closer to their nearest successful trajectories than are the true failure trajectories. Additionally, the onset of failure with respect to the onset of perturbation for false successes averaged 9.47% earlier than in the true failures. Note that it is plausible that individuals switched control strategies for some subset of the trials (e.g. from Quasi-Static to natural), which could affect the accuracy of the strategy-specific Stability Basins. Because we only characterize trials that are sits or steps as failures, it is possible that an individual switched strategies to successfully stand but would not have reached standing with their original strategy.

We compared the accuracy of Stability Basins computed using the Input Bounds controller to Stability Basins formed using other methods (table 4). The Stability Basins formed using all three controllers as well as the naive method for subject ID 1's natural Sit-to-Stand strategy are shown in figure 9.

**Table 3.** The accuracy of Stability Basins generated using the Input Bounds controller model are reported for cable pull trials at three different force levels of the applied perturbation. The results are aggregated across subjects and the three tested Sit-to-Stand control strategies.

| cable pull force level | successful trials | step trials | sit trials |
|---|---|---|---|
| low | 170/176—96.59% | 15/15—100% | 4/5—80.00% |
| medium | 125/132—94.70% | 34/38—89.47% | 24/27—88.89% |
| high | 80/85—94.12% | 61/68—89.71% | 43/45—95.56% |

Stability Basins formed using the LQR controller proposed in previous work [30] correctly predict 12.27% of successful trials. Stability Basins formed using a traditional FF+FB controller correctly predict the outcome of 46.80% of successful trials. Finally, a naive method for estimating stability yields an accuracy of 16.67% for successful trials. Though each of the LQR, FF+FB and naive methods yield nearly 100% prediction accuracy for trials where failure occurred, the low successful trial prediction rates mean that few trials remain within each Stability Basin, implying a severe underapproximation of the true stable region. We also give the false successful prediction and false failure prediction rates of each method in table 4, and display these rates for each strategy in figure 10. Each of the LQR, FF+FB and naive approaches for computing Stability Basins has a high false failure prediction rate, further showing that each underestimates the size of the true stable region.

Because the LQR, FF+FB and naive methods all appear to underestimate the true stable region, we tested the accuracy of the Stability Basins generated by these methods when they are dilated by 5%. This was accomplished by multiplying each of their Stability Basins' zonotope generators by 1.05, as in the target set generation. The classification rates for these methods do improve, though not substantially. As seen in table 5, the successful trial classification rate rises around 3% for the LQR controller, around 6% for the FF+FB controller and around 5% for the naive method. To determine what would happen if we extended this analysis further, we also evaluated the Stability Basins when the basins are dilated by 25%. This further improves the successful trial classification rate of the LQR, FF+FB and naive methods. However, at this dilation level, the FF+FB has a similar unsuccessful trial classification rate as the Input Bounds controller (89.41% compared to 91.41%), but a 41% higher false step/sit prediction rate (56.83% compared to 15.02%).

This implies that the Input Bounds controller still yields the most accurate estimate of stability. The Input Bounds controller maintains a high accuracy for successful trials, while minimizing incorrect step and sit predictions. It requires no dilation to achieve this performance. The higher false step/sit prediction rates for the LQR and FF+FB controllers despite dilation indicates that the shape of the Stability Basin constructed using the Input Bounds controller is a better approximation of the true stable region.

## 4. Discussion

In theory, stability can be directly characterized by perturbing a subject in an infinite number of ways and drawing a volume around the observed successful trials. Because this approach is practically infeasible, several model-based metrics have been developed for assessing dynamic stability. Many, such as Floquet Multipliers and Lyapunov Exponents, are limited to analysing small perturbations during periodic motion. Only a few metrics assess the largest perturbations that can be withstood during motion [21], or the stability of aperiodic motion. In this area, Fujimoto & Chou [45] constructed estimates of the stable region for Sit-to-Stand at a single time based on the extrapolated centre of mass concept. We build on these methods by leveraging novel data-driven controller models and reachability analysis to estimate stability over the course of the Sit-to-Stand motion. By performing a perturbative experiment, we demonstrate that our Stability Basin-based approach accurately predicts stability across a diverse set of body morphologies, perturbations and Sit-to-Stand control strategies.

Our data-driven controller models consist of a feed-forward ('predictive') and a linear feedback ('reactive') component. Although there is evidence that humans employ feed-forward and feedback terms for control [37], the exact form of these controllers remains unclear. Linear feedback applied about the COM has been sufficient to explain observed walking [46] and standing [47] behaviour. However, we found the LQR controller model proposed by Shia *et al.* [30] as well as the classical FF+FB controller model had poor prediction rates for successful trials, implying that they both

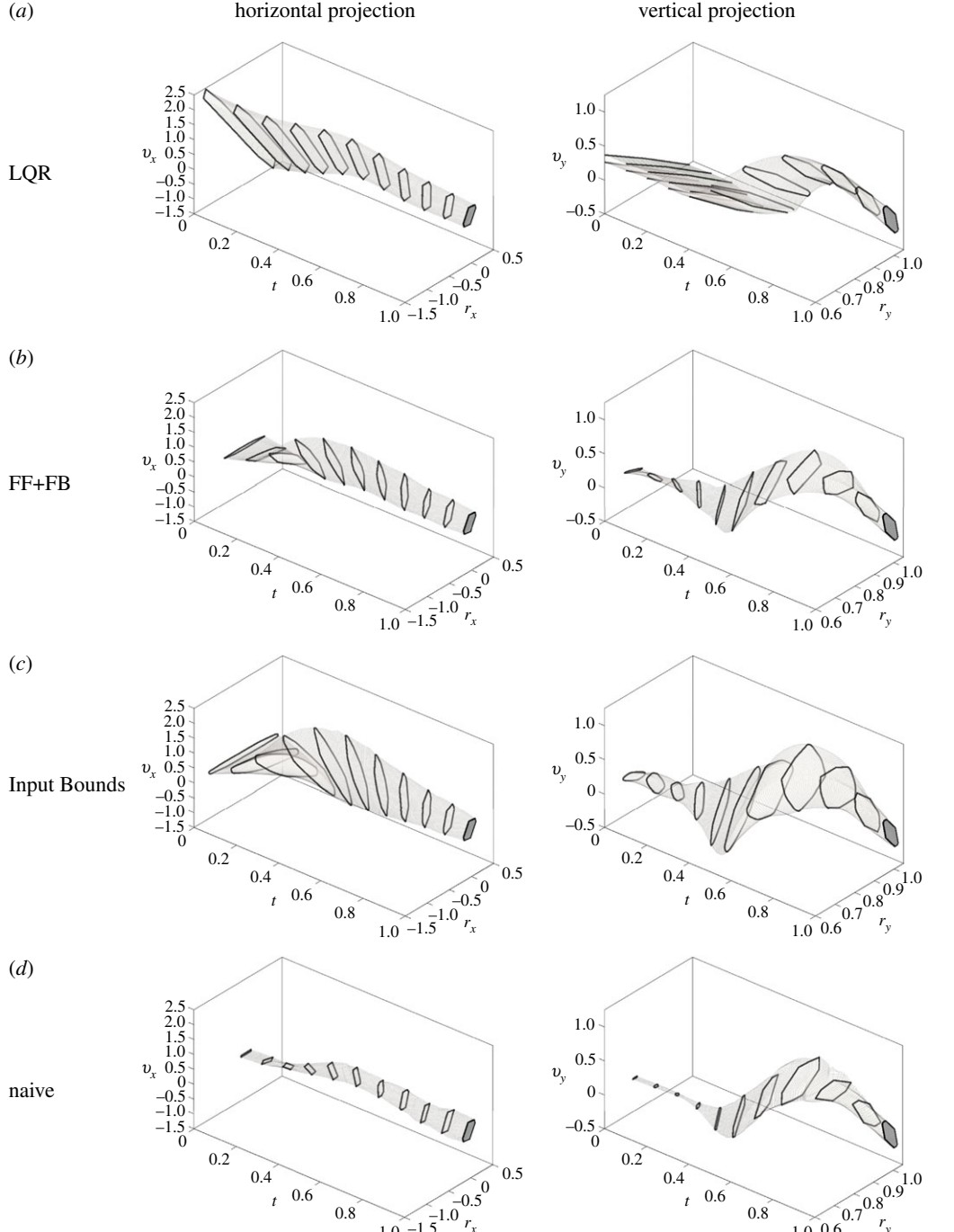

**Figure 9.** The Stability Basins formed for subject ID 1's natural Sit-to-Stand strategy using the LQR controller model (*a*), FF+FB controller model (*b*), Input Bounds controller model (*c*), and the naive method (*d*) are shown. The horizontal and vertical projections of the Stability Basins are represented as the regions encapsulated by the light grey borders, where black outlines are used to emphasize the changes in shape of cross-sections over time. The projection of the target set $X_T$ is shown as the dark grey region on the right side of each plot.

underestimate the true stable region of movement. In contrast, our Input Bounds controller model draws from a set of bounded inputs, plus a linear feedback term, that encapsulates inputs from observed successful trials, resulting in improved performance over models that return a single input.

The poor prediction rates of the LQR and FF+FB controller models for successful trials may be explained by considering the inputs that cause trajectories to exit the Stability Basin. A trajectory's input not following a given controller model is not a sufficient condition for the trajectory to exit the Stability Basin generated for that controller model. However, it is a necessary condition: if a trajectory

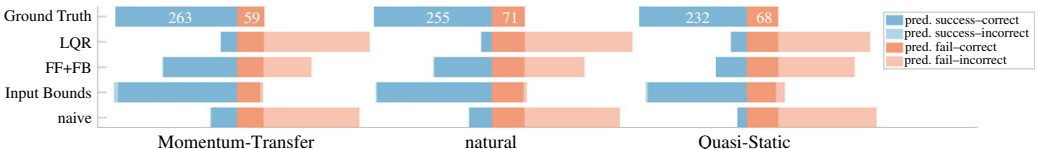

**Figure 10.** This figure compares the predictive accuracy of the SBs formed using the Input Bounds controller to three other methods. Results are presented for each Sit-to-Stand strategy, aggregated across subjects. The Input Bounds controller's predictions for both successes and failures are near the ground truth, while the three other methods predict many more failures than were experimentally observed.

**Table 4.** The accuracy of the Stability Basins formed using the Input Bounds controller model were compared to Stability Basins formed using other methods. The results of the comparison are reported below, and are aggregated across subjects and Sit-to-Stand strategies. Electronic supplementary material that reports these results for individual subjects and Sit-to-Stand strategies is available online.

| basin type | successful trials | step/sit trials | false successful predictions | false step/sit predictions |
|---|---|---|---|---|
| LQR | 92/750—12.27% | 197/198—99.49% | 1/93—1.08% | 658/855—76.96% |
| FF+FB | 351/750—46.80% | 195/198—98.48% | 3/354—0.85% | 399/594—67.17% |
| Input Bounds | 718/750—95.73% | 181/198—91.41% | 17/735—2.31% | 32/213—15.02% |
| naive | 125/750—16.67% | 195/198—98.48% | 3/128—2.34% | 625/820—76.21% |

**Table 5.** The accuracy of the Stability Basins for each method are compared when dilating the LQR, FF+FB and naive Stability Basins. The results are aggregated across subjects and STS strategies.

| basin type | successful trials | step/sit trials | false successful pred. | false step/sit pred. |
|---|---|---|---|---|
| Input Bounds | 718/750—95.73% | 181/198—91.41% | 17/735—2.31% | 32/213—15.02% |
| LQR (no dilation) | 92/750—12.27% | 197/198—99.49% | 1/93—1.08% | 658/855—76.96% |
| FF+FB (no dilation) | 351/750—46.80% | 195/198—98.48% | 3/354—0.85% | 399/594—67.17% |
| naive (no dilation) | 125/750—16.67% | 195/198—98.48% | 3/128—2.34% | 625/820—76.21% |
| LQR (5% dilation) | 113/750—15.07% | 194/198—97.98% | 4/117—3.42% | 637/831—76.65% |
| FF+FB (5% dilation) | 390/750—52.00% | 192/198—96.97% | 6/396—1.52% | 360/552—65.22% |
| naive (5% dilation) | 159/750—21.20% | 195/198—98.48% | 3/162—1.85% | 591/786—75.19% |
| LQR (25% dilation) | 160/750—21.33% | 188/198—94.95% | 10/170—5.88% | 590/778—75.84% |
| FF+FB (25% dilation) | 517/750—68.93% | 177/198—89.39% | 21/538—3.90% | 233/410—56.83% |
| naive (25% dilation) | 292/750—38.93% | 189/198—95.45% | 9/301—2.99% | 458/647—70.79% |

exits the Stability Basin, then the trajectory's input must not have exactly followed the controller model at some point. Therefore, the accuracy of the Stability Basins strongly depends on how well each controller model fits the observed data. As can be seen in figure 3, the Input Bounds controller accounts for a wider range of inputs than the LQR and FF+FB controllers, and is able to encapsulate the variability present in the observed data. This in turn creates a Stability Basin which better matches the size and shape of the true stable region. This improvement in the accuracy of stability estimation is an important step towards clinical deployment.

In addition to increased accuracy, the computationally-tractable, individualized and time-varying aspects of the Stability Basin approach make it well suited for widespread deployment and integration with existing clinical methods. The rapid and automated computation of Stability Basins from kinematics alone minimizes the temporal and financial investment in performing this analysis,

especially if coupled with computer vision techniques for pose estimation from digital cameras [48]. This can enable longitudinal studies examining how stability changes over time, and whether a smaller Stability Basin is correlated with fall risk, to be performed both accurately and cheaply. Although we make use of perturbative data for computing Stability Basins here, one advantage of our proposed method is that it only relies on data from successful trials, and does not require a subject to be perturbed to failure. Thus, the Stability Basin method shows promise for characterizing the stability of subjects already at a high risk of falling or injury, such as elderly or rehabilitating subjects. Additionally, the TIPM and Input Bounds models are computed for each individual, enabling examination of a wide range of body morphologies and Sit-to-Stand control strategies. Furthermore, stability throughout the duration of a task can be studied by measuring the size and location of cross-sections of the Stability Basin at particular times. For example, cross-sections of the Stability Basin that examine COM position at seat-off reveal the viable starting postures for each control strategy [30,45]. In particular, Quasi-Static Stability Basins were previously shown to include COM positions near one's feet at seat-off, whereas Momentum-Transfer Stability Basins can extend much further. We propose that in longitudinal studies with periodic data collection, sudden changes in Stability Basin volume overall or in a portion of the movement can indicate how injury or age contribute to instability. By combining the Stability Basin approach with other clinical techniques, such as electromyography, clinicians can identify muscles associated with unstable portions of motion to target preventative and rehabilitative care. Such detailed analyses can provide deeper insight into the mechanisms underlying motor control.

Our approach can also be used in real time to inform controller design for wearable robotic devices because it provides time-varying boundaries on the region inside which an individual is stable. Determining the state at which a robotic system is unable to recover from a perturbation is key to designing dynamically stable controllers [49–51]. If a prosthetic device or exoskeleton detects that its user has exited the Stability Basin and become unstable, it can adapt its control scheme to aid in recovery. Furthermore, just as current tuning of controllers for prosthetic devices and exoskeletons focuses on minimizing metabolic cost, [52,53], wearable robotic controllers can be optimized to increase stability by maximizing the volume of the Stability Basins for the combined robot-human system. Thus, the high time-varying accuracy and low investment in computation demonstrate that the Stability Basin approach has the potential to increase access of personalized preventative and rehabilitative mobility care and inform the design of robotic systems for stable movement.

Ethics. The experimental protocol was approved by the University of Michigan Health Sciences and Behavioral Sciences Institutional Review Board, eResearch ID: HUM00020554.

Data accessibility. Code and Sit-to-Stand data may be found here: https://github.com/pdholmes/STS_SB. Alternatively, code and Sit-to-Stand data is available at this permanent repository: Holmes, P. Perturbative Sit-to-Stand Experiment Dataset and Stability Basin Code [Data set]. University of Michigan - Deep Blue. https://doi.org/10.7302/mhjr-k798.

Authors' contributions. P.D.H. conducted the experiment, analysed the data, wrote Stability Basin computation and validation code and drafted the manuscript; S.M.D. analysed the data, wrote Stability Basin computation code and drafted the manuscript; X.-Y.F. constructed perturbation equipment, conducted the experiment and edited the manuscript; T.Y.M. conducted the experiment, supervised data analysis and edited the manuscript; R.V. conceived and directed the study, and edited the manuscript.

Competing interests. We declare we have no competing interests.

Funding. This work is supported by the National Science Foundation, under CAREER Award 1751093 and Graduate Research Fellowship Program grant no. 1256260 DGE.

Acknowledgements. We thank Art Kuo for helping to collect this dataset, and Victor Shia for his code and advice. We thank the 11 anonymous subjects for their participation in the study.

# Appendix A. Generating cable pull perturbations

Two high-torque motors were attached to cable-pulley systems and placed in front of and behind the test platform. The cables were attached to a belt around the subject's waist, and the height of the pulleys were adjusted such that the cables were horizontal when the subject was standing. The pulleys were located approximately 3–4 feet in front of and behind the subjects. The cables were only attached for the cable pull condition, and detached for the nominal and foot-shift conditions.

A custom-written LabVIEW [54] program was used to control the motor torques. During the experiment, a low torque was constantly commanded in both motors to keep the cables from going slack. The torque level was balanced between the anterior and posterior cables to minimize force bias felt by the subject. In particular, both the front and back perturbation motors were commanded to

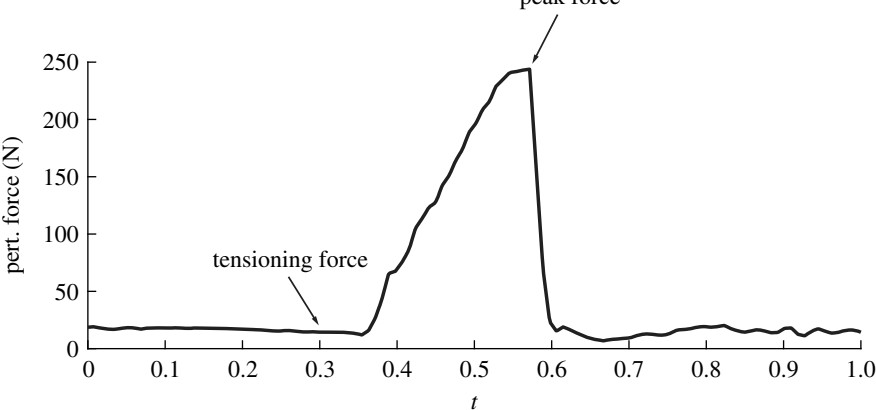

**Figure 11.** A typical perturbation time series is shown in this figure. This particular perturbation was a forwards perturbation at the medium force level. A baseline tensioning force of 19.6 N is commanded, until the start of the perturbation. The force quickly ramps up to a specified peak, then back down to the baseline tensioning force.

maintain a baseline tensioning force of 19.6 N, so that the net force on the subject was approximately 0 N when no active perturbation was applied. Force sensors on the cables were synchronized with motion capture data, so that the timing of the perturbation application in relation to kinematic data was known. Perturbations were manually activated by the toggling of a handheld switch. The experiment operator applied perturbations to the subject at variable times following seatoff to gather a range of recovery responses over the course of Sit-to-Stand. Perturbations were generally applied at around 50% of the motion, shown in the Pert. Onset column of table 1.

Perturbations were active over a 250 ms period, during which the applied force would ramp up to a specified peak, and then quickly ramp down to the baseline tensioning force, as shown in figure 11. The peak force to be applied was determined before the experiment began. Specifically, three peak force levels—low, medium and high—were calibrated to each subject. The low force level was designed to rarely induce stepping or sitting during Sit-to-Stand, while the high force level was designed to induce stepping or sitting approximately half of the time. Typical values for the peak forces applied at each force level, normalized by each subject's body weight, are shown in figure 12. On average, peak forces were around 22% of body weight at the low force level, 32% at the medium force level and 37% at the high force level.

Stepping and sitting are more formally defined in §2.5.1, and we report the number of steps and sits observed for each force level in §3. These force levels were roughly chosen based on the subject's height and weight, and adjusted manually during a pre-experiment test session. Six trials were taken at each force level for each Sit-to-Stand strategy, with three pulling forwards and three pulling backwards, in random order.

## Appendix B. Determining start and end of Sit-to-Stand

To segment the continuous kinematic Sit-to-Stand data into individual trials, an appropriate start and end time for each trial must be chosen. A consistent segmentation method is imperative for making comparisons across trials. Ideally, the start and end of each trial is chosen so that important kinematic features of Sit-to-Stand are aligned.

We developed the following segmentation procedure, displayed in figure 13, which is designed to align the peak horizontal accelerations of Sit-to-Stand trials while being robust to differences in trial duration. Portions of the procedure use a root-mean-square error (RMSE), defined as:

$$\text{RMSE}(\hat{y}, y) = \sqrt{\frac{\sum_{i=1}^{N}(\hat{y}_i - y_i)^2}{N}}, \qquad (\text{B 1})$$

where $\hat{y} = \{\hat{y}_i\}_{i=1}^{N}$, $\hat{y}_i \in \mathbb{R}$ and $y = \{y_i\}_{i=1}^{N}$, $y_i \in \mathbb{R}$ are two sequences defined by sampling two different trajectories at $N$ times.

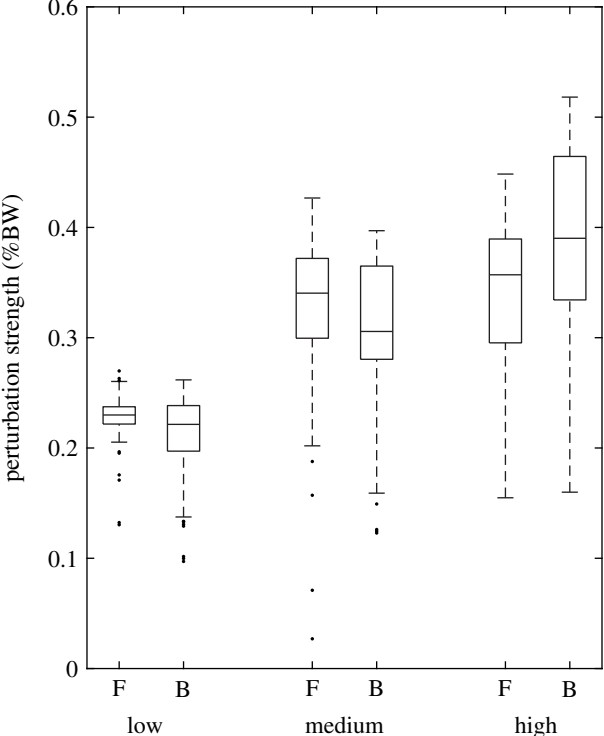

**Figure 12.** This figure shows typical peak perturbative force values (normalized by subject body weight) for the forwards (F) and backwards (B) perturbations applied at the low, medium and high force levels. These peak forces were averaged across subjects and Sit-to-Stand strategies to produce this plot. For each force level, the central mark indicates the median peak force. The bottom and top edges of the box indicate the the 25th and 75th percentiles, respectively. The whiskers extend to the most extreme forces not considered outliers, and the outliers are plotted individually.

The procedure generates an average nominal trial from the subject's five nominal trials, and then uses it as a template to segment the rest of the trials. We segment the natural and Momentum-Transfer strategies' trials using the following procedure, but use a separate procedure for Quasi-Static trials, detailed later on in this section.

1. Manually segment each trial, starting several seconds before the subject begins to stand and ending several seconds after they have reached standing.
2. To create an average nominal trial:
   (a) Take the manual segmentations of the subject's five nominal trials, and align their COM position trajectories such that the times at which the peak horizontal COM velocities occur are coincident.
   (b) Average the subject's five nominal trials together to form a single average nominal COM position trajectory.
   (c) Numerically differentiate the average nominal COM position trajectories to obtain velocity and acceleration trajectories.
   (d) Choose the start time ($\tau = 0$) of the average nominal trajectory as when the horizontal COM acceleration first exceeds 20% of its max. Choose the end time ($\tau = T_{nom}$) as when the vertical COM position first exceeds 99% of its max.
   (e) The average nominal trajectory is now defined over $[0, T_{nom}]$. Follow the scaling laws in §2.2.2 so that it is defined over $[0, 1]$.
3. To segment each trial:
   (a) Iterate over times $T \in [T_{min}, T_{max}]$, where the interval is discretized into 100 samples. Let $T_{min} = 0.75$ s and $T_{max} = 1.5 T_{nom}$.
   (b) Rescale the average nominal trajectory to be defined over $[0, T]$.
   (c) Automatically align the scaled average nominal trial and the current trial so that the time at which the peak horizontal COM accelerations occur are coincident.
   (d) The trajectories overlap in a window of length $T$ seconds. Compute the RMSE between the horizontal COM acceleration trajectories over this window.

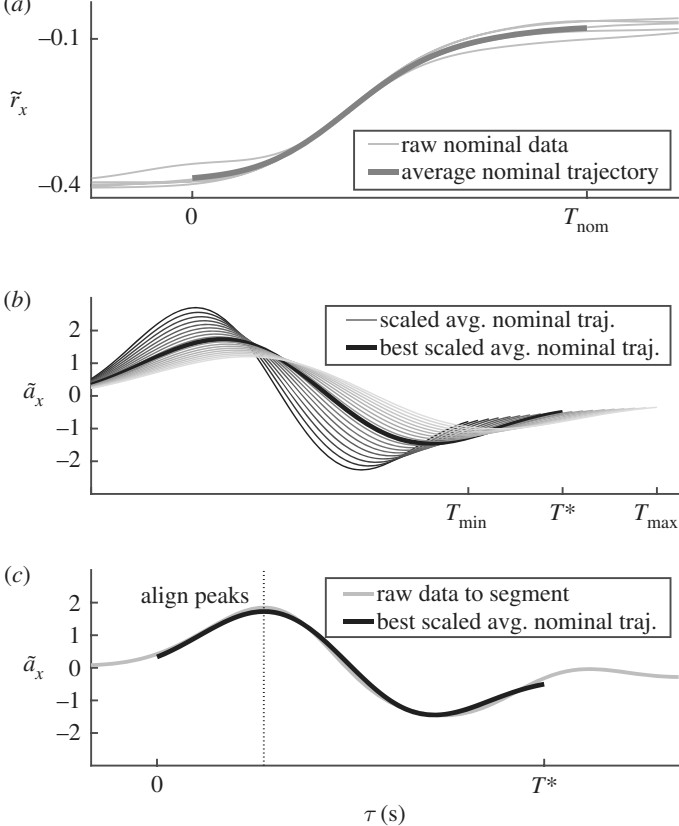

**Figure 13.** This figure illustrates the segmentation procedure. Step 2(b) of the procedure is depicted in (a). We take the subject's five nominal trials, shown in light grey, and average them to form a single average nominal trajectory, shown as the thick darker grey line. We choose a start and end time [0, $T_{nom}$] for the average nominal trajectory as in Step 2(d). Step 3(a–b) of the procedure is depicted in (b). We obtain the horizontal acceleration trajectory of the average nominal trajectory by twice differentiating the position trajectory shown in (a). We then iterate over time scales in [$T_{min}$, $T_{max}$], scaling the trajectory according to §2.2.2. These scaled trajectories are shown in shades of grey. An example trajectory corresponding to time $T^*$ is shown as the thick black line. Step 3(c–e) of the procedure is depicted in (c). The horizontal acceleration trajectory of a trial to be segmented is shown as the thin light grey line. Of the scaled trajectories in (b), the one corresponding to time $T^*$, shown as the thick black line, matches the best. The start and end times of the trial to be segmented are then determined by the endpoints of the overlaid black line.

(e) Find $T$ that minimizes this RMSE (to accurately segment cable pull trials despite the effects of perturbation, we only minimize the pre-perturbation RMSE for cable pull trials).

Because the Quasi-Static strategy uses minimal forward momentum, it lacks a distinct peak in its horizontal COM acceleration trajectories when compared to the natural and Momentum-Transfer strategies. This is shown in figure 7. Therefore, we developed a separate segmentation procedure for the Quasi-Static strategy, where Steps 1 and 2 remain the same as before:

4. To segment each Quasi-Static trial:

   (a) Iterate over times $T \in [T_{min}, T_{max}]$, where the interval is discretized into 100 samples. Let $T_{min} = 0.75$ s and $T_{max} = 1.5T_{nom}$.
   (b) Rescale the average nominal trajectory to be defined over [0, $T$].
   (c) Manually align the scaled average nominal trial and the current trial so that the time at which the peak vertical COM velocities occur are coincident.
   (d) The trajectories overlap in a window of length $T$ seconds. Manually choose $T$ that minimizes the RMSE between the scaled average nominal vertical COM velocity trajectory and the current trial's COM velocity trajectory. (Prioritize the pre-perturbation difference for cable pull trials, to accurately segment cable pull trials despite the effects of perturbation.)

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
