## [Reviewer comments · Royal Society Open Science]

Review History

RSOS-191410.R0 (Original submission)

Review form: Reviewer 1 (Sjoerd Bruijn)

Is the manuscript scientifically sound in its present form?

Yes

Are the interpretations and conclusions justified by the results?

Yes

Is the language acceptable?

Yes

Do you have any ethical concerns with this paper?

No

Have you any concerns about statistical analyses in this paper?

No

Recommendation?

Accept with minor revision (please list in comments)

Comments to the Author(s)

Review: RSOS-191410

Title: Characterizing the limits of human stability during motion: perturbative experiment validates a model-based approach for the Sit-to-Stand task

This manuscript describes a study undertaken to assess the validity of basin of stability as obtained from a sit-to-stand experiment. To this aim, 11 subjects are asked to perform sit-to-stand movements, and perturbations are used. During some trials, subjects could not perform a successful sit-to-stand (due to perturbations). Results showed that basin of stability calculations could accurately predict which trials would be unsuccessful.

All in all, this is an interesting study, and the results seem promising. It is also written in a very readable way (except maybe the extensive use of abbreviations, which does not really help the reader). It could be because of my own limited background with the subject, but some of the methodological subtleties were less clear, and could thus be improved upon. I will try to outline these below.

(Somewhat) major comments:

- 1) It is unclear to me how the BFF+FB controller differs from the other controllers, and I feel it would be good if the authors spend a few more words on this. For the LQR controller, it seems that one obvious difference is that it only uses non-perturbed trial data (am I correct here?). For the other controller, the difference is less clear to me. In order to appreciate the manuscript to its fullest, it would be good if the authors spend some more words on this.
- 2) Also, if my reading is correct and the BFF+FB controller used the perturbed trials (but only the parts which were still successful), while the LQR does not, what does that mean? Ideally, we would want to use these kind of measures to obtain a measure of stability from unperturbed movements, right?
- 3) Discussion, page 15, lines 6-7: "Furthermore, stability throughout a task can be studied by measuring the size and location of cross-sections of the SB at particular times."; I wonder whether this would be informative? Does the size of the basin necessarily say how likely people are to get outside of the basin? I think most likely not?

Minor comments:

- 1) As there is no word limit, I suggest the authors remove most (if not all) abbreviations. These are mostly confusing for readers, especially for a broader audience (such as to be expected in this journal).
- 2) Abstract: "11 person perturbative STS experiment" sounds a bit odd? Consider changing to something like: Perturbation experiment, in which 11 subjects were included? (see also intro, page 2 line 21, and possibly other places)
- 3) The definition of stability is only given in the methods, first few sentences. It might be good to have that earlier in the intro?
- 4) Page 4, line 20: "We define the origin as the initial position of the ball of the subject's foot". However, in the figure, this appears to be the ankle? And how exactly is "the ball of the foot" defined?
- 5) Caption of figure 4: it may be good to change to "how X_t (the standing set)...."

Review form: Reviewer 2

Is the manuscript scientifically sound in its present form?

No

Are the interpretations and conclusions justified by the results?

Yes

Is the language acceptable?

Yes

Do you have any ethical concerns with this paper?

No

Have you any concerns about statistical analyses in this paper?

No

Recommendation?

Major revision is needed (please make suggestions in comments)

Comments to the Author(s)

The paper presents an interesting modelling approach for characterising stability during a sit-to-stand task using stability basins, validated using experiments. In previous work by Shia et al 2018, the calculation of the stability basin depends on the specific controller which a subject uses. In this paper, this assumption is loosened, so that only the bounds on the control input are required to calculate stability basins. The authors validate the model using data from waist-pull perturbations during a sit-to-stand task. During most of the failed trials (in which the subject had to sit back down or take a step due because of the perturbation), the subject's trajectory left the stability basin before the failure occurred. Inversely, during most of the successful trials, the subject's trajectory remained in the stability basin.

I have major concerns about the controller modelling assumptions and the experimental comparison of the different controllers. Moreover, the Methods section is very difficult to read. I also have additional comments, listed below.

Controller modelling assumptions

In Shia et al 2018, the control input is assumed to follow an open-loop control with feedback control around a nominal trajectory. This paper considers a more general control input, which is bounded for each time and state. From a biomechanical perspective, it makes sense that the ground reaction force is limited, and that the limits depend on the system state. However, I see no reason why the bounds should also depend on the normalised time, please justify this assumption.

The authors further assume that the bounds apply to the normalised control input, rather than the actual ground reaction force. I see however no reason why a subject should be able to produce larger forces in trials with a shorter duration.

In equations (6) and (7), I think it is misleading to call the first term "feed-forward" and the second term "feed-back". Presumably, after a perturbation the subjects will adjust the first term (within the bounds) to compensate for the perturbation, so it is not entirely feedforward.

Why are the bounds only fit on successful trials rather than on all trials? In unsuccessful trials, does the control input remain within the bounds determined from successful trials?

Why are the bounds (and the standing set) fit separately for Quasi-static, Natural, and Momentum transfer trials? I expect the same biomechanical constraints to apply for each of these tasks, resulting in the same bounds on the ground reaction force.

Experimental comparison of the controllers

Why do the alternative controllers have such poor prediction rates for successful trials? If the stability basins for the LQR and FF+FB controllers are extended by 5% (similar to standing set), does this provide a better classification?

LQR controller

In the Discussion, the authors mention that “the LQR controller proposed by Shia et al. [...] is not trained on any of the observed perturbed trials” (p. 14, lines 46 – 47). If this is also the case in your paper, then it is an unfair comparison. In the Methods 2.5.1, the authors say that, to generate the feedback matrix for the LQR, they use “weighting matrices [which] were found empirically to produce the best results”. What does “best” mean? Does it provide the best classification performance for cable-pull perturbation trials?

FF + FB controller

Were the terms $ff(t)$ and $K(t)$ fit to the data from all successful trials, or only nominal and foot-shift trials? Are the values of $K(t)$ obtained from fitting the FF+FB controller similar to those obtained from fitting the BFF+FB controller?

BFF + FB controller

If the stability basins are calculated only from nominal and foot-shift trials, does this accurately predict success and failure in cable-pull perturbation trials?

Organisation of the Methods

All of the information relevant to the experiment and its analysis should be in a single section (presumably 2.1 Perturbative STS experiment):

2.1.3 Classifying trials as successful/unsuccessful (p.5 lines 2 to 8):

It is inconvenient for stepping and sitting onset to be defined several pages later (p.9 lines 41 to 47); the definition should be moved to Section 2.1.3. The classification of foot-shift trials (found p.11 lines 43 to 46) should also be moved here, and it should be clarified: when a subject is unable to initiate the STS movement, or violates the parameters of acceptable trials, then the trial should not be considered successful. Please replace “this scheme can classify” (p.5 line 7) by “this scheme classifies”.

The following information belongs to the Methods rather than the Results:

- P. 11 lines 32 – 34 “We collected 948 trials from 11 participants (three female and eight male; ages 18-32; height 1.70 +/- 0.12 m; body mass 65.4 +/- 10.2 kg), including 213 unperturbed, 194 foot-shifted, and 591 cable pull trials. Each subject gave their informed written consent, and had no physical or balance disorders which could affect their ability to perform STS.”

- p. 11 lines 51 – 52 “SBs were generated on a laptop computer with a 2.7 GHz Intel Core i7 processor. SBs computed using the BFF+FB controller take 0:76+/-0:014 seconds to compute.”

The three different controllers (BFF+FB, LQR, FF+FB) should be presented in a single section, to highlight their differences. Moreover, the modelling assumptions for each controller need to be justified. Figure 3 should be extended to illustrate the LQR and FF+FB controllers, similar to panel (b). It would be helpful to color-code the dots in panel (b) to show those that correspond to nominal, foot-shift, successful perturbation and unsuccessful perturbation trials. It would also be useful to indicate which of these trials were used to fit the three different controllers (as it seems the controllers were not all fit on the same subset of trials).

The procedure for computing stability basins should be briefly described in comprehensible terms and explained through a sketch, but most of the mathematical definitions as well as section 2.4.2 should be moved to the Appendix.

The procedure for validating the stability basins is difficult to follow. Could you please provide a graphical representation indicating which trials are used for which steps of the procedure, for each of the strategies and the controllers? For example, it is not clear to me in which step the foot-shift trials are analysed.

Additional comments

In the abstract, the second sentence is about identifying people and the third sentence is about identifying perturbations. The articulation between the two is not clear.

The expression "an 11-person perturbative STS experiment" (p. 2 line 29 and p.3 line 21) is confusing. Please reformulate, for example: "a perturbative STS experiment with 11 subjects" or "11 subjects participated in a perturbative STS experiment".

Switch in control strategy:

The authors first say a switch in control strategy is "indicated by a step or sit" (Abstract p.2 lines 30 -31), then p.9 line 35 "Previously in Sec. 2.1.3 we defined strategy switches as stepping or sitting back down". However, in the Methods, the switch in control strategy is presented as an alternative to stepping or sitting: p.4 lines 5 - 6 "stepping, sitting back down, or switching control strategy"; p.4 lines 9 - 10 "take a step, sit back down, or switch control strategy".

Moreover, in the rest of the text, control strategies actually refer to "natural", "momentum transfer" and "quasi-static" (ex: p.4 lines 26 - 27).

This is very confusing.

2.1.1 Experimental protocol:

"Foot shift (FS) perturbations" (p.4, line 32): these are not really perturbed trials, so to avoid confusion they should perhaps simply be called "foot shift trials". How many trials were performed at each foot position? Which strategy were the subjects instructed to use?

2.2.1 p.5 line 20: "We define the origin as the initial position of the ball of the subject's foot": is this indicated by a kinematic marker on a given anatomical landmark?

2.3 Computing Stability Basins

Definition 1: starting from the stability basin, the system will not necessarily arrive at the standing set, for example if the control inputs are not well chosen.

How many generators did you use for the standing set and the stability basins (i.e. what is the value of p)?

2.3.3 p. 9 lines 28 - 29 "CORA therefore assumes the input at a given time and state can take any value within the input bounds to create the SB." How is this modified to calculate the Stability Basins for the LQR and FF+FB?

2.4.3 p.10 lines 25 - 26: In the leave-one-out procedure, why is the trial not also removed from the calculation of the standing set?

2.5.1 p.10 lines 40 to 42: the way in which the feedback matrix is determined is very obscure. What are the matrices Q and R and what does "best results" mean?

3. Results

p. 11 lines 34 – 35 “The subjects performed STS trajectories with generally the same characteristics as the (different) subjects examined by Shia”: what are these characteristics?

p.11 lines 41 – 42 “The SB shape and volume are highly dependent on the kinematic and temporal characteristics corresponding to the strategy selected, as in previous work”: how do the shape and volume change with strategy?

The units used to report distances are inconsistent: heights are reported in m, accelerations in m/s^2 , but foot positions in inches. Please consistently use the International System of Units and convert foot positions to m (or mm).

p.11 lines 55 – 56: how long in advance are failures predicted?

Figure 7: please indicate in the legend what the lines, boxes and whiskers correspond to (i.e. means, medians, percentiles?)

Figure 8: Please describe the figure in the main text. The light grey shading is too difficult to see. The names of the panels (a – Successful, b - Step, c – Sit) should be written horizontally. Please also illustrate a successful perturbation trial. The time during which the perturbation is active is indicated as a thick dashed line in Figure 1 and as a thick non-dashed line in Figure 8. Please choose a consistent representation.

Figure 9: the shades of blue and red are darker in the ground truth than in the other lines. The shades for incorrect predictions are too light.

Discussion

p. 15 line 2: the method nevertheless requires perturbation trials in order to determine the stability basin, which may not be convenient to administer in a clinical setting. Moreover, the stability basin does not indicate whether or not a subject will be able to successfully resist a given perturbation. The classification of successful and unsuccessful perturbation trials relies on the motion capture data after the perturbation.

Appendix A

p. 18 lines 12 - 13: Does the range of perturbation onset timings cover the whole duration of the STS trials?

p. 18 lines 14 - 15: Is the baseline force level at the end of the perturbation non-zero? Is the baseline force level before the perturbation zero?

p. 18 line 16: Please report the range of peak forces across subjects, normalised by subject weight, for each of the three force levels. It would also be helpful to include a figure with the perturbation time course.

Appendix B p. 19 line 44: is the alignment done manually?

Decision letter (RSOS-191410.R0)

04-Oct-2019

Dear Mr Holmes,

The editors assigned to your paper ("Characterizing the limits of human stability during motion: perturbative experiment validates a model-based approach for the Sit-to-Stand task") have now

received comments from reviewers. We would like you to revise your paper in accordance with the referee and Associate Editor suggestions which can be found below (not including confidential reports to the Editor). Please note this decision does not guarantee eventual acceptance.

Please submit a copy of your revised paper before 27-Oct-2019. Please note that the revision deadline will expire at 00.00am on this date. If we do not hear from you within this time then it will be assumed that the paper has been withdrawn. In exceptional circumstances, extensions may be possible if agreed with the Editorial Office in advance. We do not allow multiple rounds of revision so we urge you to make every effort to fully address all of the comments at this stage. If deemed necessary by the Editors, your manuscript will be sent back to one or more of the original reviewers for assessment. If the original reviewers are not available, we may invite new reviewers.

- Data accessibility

If you wish to submit your supporting data or code to Dryad (<http://datadryad.org/>), or modify your current submission to dryad, please use the following link:
<http://datadryad.org/submit?journalID=RSOS&manu=RSOS-191410>

- Competing interests

- Authors' contributions

- Acknowledgements

- Funding statement

on behalf of Dr Monica Daley (Associate Editor) and Kevin Padian (Subject Editor)
openscience@royalsociety.org

Editor comments:

Thanks very much for your submission. Overall the reviewer comments suggest that the approach makes a potentially valid and useful contribution to the literature, but the paper requires substantial revisions to make the reasoning and interpretation of the modeling approaches clearer. Best success for your revisions.

Reviewers' Comments to Author:

Reviewer: 1

Comments to the Author(s)

Review: RSOS-191410

Title: Characterizing the limits of human stability during motion: perturbative experiment validates a model-based approach for the Sit-to-Stand task

This manuscript describes a study undertaken to assess the validity of basin of stability as obtained from a sit-to-stand experiment. To this aim, 11 subjects are asked to perform sit-to-stand movements, and perturbations are used. During some trials, subjects could not perform a successful sit-to-stand (due to perturbations). Results showed that basin of stability calculations could accurately predict which trials would be unsuccessful.

All in all, this is an interesting study, and the results seem promising. It is also written in a very readable way (except maybe the extensive use of abbreviations, which does not really help the reader). It could be because of my own limited background with the subject, but some of the methodological subtleties were less clear, and could thus be improved upon. I will try to outline these below.

(Somewhat) major comments:

- 1) It is unclear to me how the BFF+FB controller differs from the other controllers, and I feel it would be good if the authors spend a few more words on this. For the LQR controller, it seems that one obvious difference is that it only uses non-perturbed trial data (am I correct here?). For the other controller, the difference is less clear to me. In order to appreciate the manuscript to its fullest, it would be good if the authors spend some more words on this.
- 2) Also, if my reading is correct and the BFF+FB controller used the perturbed trials (but only the parts which were still successful), while the LQR does not, what does that mean? Ideally, we would want to use these kind of measures to obtain a measure of stability from unperturbed movements, right?
- 3) Discussion, page 15, lines 6-7: "Furthermore, stability throughout a task can be studied by measuring the size and location of cross-sections of the SB at particular times."; I wonder whether this would be informative? Does the size of the basin necessarily say how likely people are to get outside of the basin? I think most likely not?

Minor comments:

- 1) As there is no word limit, I suggest the authors remove most (if not all) abbreviations. These are mostly confusing for readers, especially for a broader audience (such as to be expected in this journal).
- 2) Abstract: "11 person perturbative STS experiment" sounds a bit odd? Consider changing to something like: Perturbation experiment, in which 11 subjects were included? (see also intro, page 2 line 21, and possibly other places)
- 3) The definition of stability is only given in the methods, first few sentences. It might be good to have that earlier in the intro?
- 4) Page 4, line 20: "We define the origin as the initial position of the ball of the subject's foot". However, in the figure, this appears to be the ankle? And how exactly is "the ball of the foot" defined?
- 5) Caption of figure 4: it may be good to change to "how X_t (the standing set)...."

Reviewer: 2

Comments to the Author(s)

The paper presents an interesting modelling approach for characterising stability during a sit-to-stand task using stability basins, validated using experiments. In previous work by Shia et al 2018, the calculation of the stability basin depends on the specific controller which a subject uses. In this paper, this assumption is loosened, so that only the bounds on the control input are required to calculate stability basins. The authors validate the model using data from waist-pull perturbations during a sit-to-stand task. During most of the failed trials (in which the subject had to sit back down or take a step due because of the perturbation), the subject's trajectory left the

stability basin before the failure occurred. Inversely, during most of the successful trials, the subject's trajectory remained in the stability basin.

I have major concerns about the controller modelling assumptions and the experimental comparison of the different controllers. Moreover, the Methods section is very difficult to read. I also have additional comments, listed below.

Controller modelling assumptions

In Shia et al 2018, the control input is assumed to follow an open-loop control with feedback control around a nominal trajectory. This paper considers a more general control input, which is bounded for each time and state. From a biomechanical perspective, it makes sense that the ground reaction force is limited, and that the limits depend on the system state. However, I see no reason why the bounds should also depend on the normalised time, please justify this assumption.

The authors further assume that the bounds apply to the normalised control input, rather than the actual ground reaction force. I see however no reason why a subject should be able to produce larger forces in trials with a shorter duration.

In equations (6) and (7), I think it is misleading to call the first term "feed-forward" and the second term "feed-back". Presumably, after a perturbation the subjects will adjust the first term (within the bounds) to compensate for the perturbation, so it is not entirely feedforward.

Why are the bounds only fit on successful trials rather than on all trials? In unsuccessful trials, does the control input remain within the bounds determined from successful trials?

Why are the bounds (and the standing set) fit separately for Quasi-static, Natural, and Momentum transfer trials? I expect the same biomechanical constraints to apply for each of these tasks, resulting in the same bounds on the ground reaction force.

Experimental comparison of the controllers

Why do the alternative controllers have such poor prediction rates for successful trials? If the stability basins for the LQR and FF+FB controllers are extended by 5% (similar to standing set), does this provide a better classification?

LQR controller

In the Discussion, the authors mention that "the LQR controller proposed by Shia et al. [...] is not trained on any of the observed perturbed trials" (p. 14, lines 46 - 47). If this is also the case in your paper, then it is an unfair comparison. In the Methods 2.5.1, the authors say that, to generate the feedback matrix for the LQR, they use "weighting matrices [which] were found empirically to produce the best results". What does "best" mean? Does it provide the best classification performance for cable-pull perturbation trials?

FF + FB controller

Were the terms $ff(t)$ and $K(t)$ fit to the data from all successful trials, or only nominal and foot-shift trials? Are the values of $K(t)$ obtained from fitting the FF+FB controller similar to those obtained from fitting the BFF+FB controller?

BFF + FB controller

If the stability basins are calculated only from nominal and foot-shift trials, does this accurately predict success and failure in cable-pull perturbation trials?

Organisation of the Methods

All of the information relevant to the experiment and its analysis should be in a single section (presumably 2.1 Perturbative STS experiment):

2.1.3 Classifying trials as successful/unsuccessful (p.5 lines 2 to 8):

It is inconvenient for stepping and sitting onset to be defined several pages later (p.9 lines 41 to 47); the definition should be moved to Section 2.1.3. The classification of foot-shift trials (found p.11 lines 43 to 46) should also be moved here, and it should be clarified: when a subject is unable to initiate the STS movement, or violates the parameters of acceptable trials, then the trial should not be considered successful. Please replace “this scheme can classify” (p.5 line 7) by “this scheme classifies”.

The following information belongs to the Methods rather than the Results:

- P. 11 lines 32 – 34 “We collected 948 trials from 11 participants (three female and eight male; ages 18-32; height 1.70 +/- 0.12 m; body mass 65.4 +/- 10.2 kg), including 213 unperturbed, 194 foot-shifted, and 591 cable pull trials. Each subject gave their informed written consent, and had no physical or balance disorders which could affect their ability to perform STS.”

- p. 11 lines 51 – 52 “SBs were generated on a laptop computer with a 2.7 GHz Intel Core i7 processor. SBs computed using the BFF+FB controller take 0:76+/-0:014 seconds to compute.”

The three different controllers (BFF+FB, LQR, FF+FB) should be presented in a single section, to highlight their differences. Moreover, the modelling assumptions for each controller need to be justified. Figure 3 should be extended to illustrate the LQR and FF+FB controllers, similar to panel (b). It would be helpful to color-code the dots in panel (b) to show those that correspond to nominal, foot-shift, successful perturbation and unsuccessful perturbation trials. It would also be useful to indicate which of these trials were used to fit the three different controllers (as it seems the controllers were not all fit on the same subset of trials).

The procedure for computing stability basins should be briefly described in comprehensible terms and explained through a sketch, but most of the mathematical definitions as well as section 2.4.2 should be moved to the Appendix.

The procedure for validating the stability basins is difficult to follow. Could you please provide a graphical representation indicating which trials are used for which steps of the procedure, for each of the strategies and the controllers? For example, it is not clear to me in which step the foot-shift trials are analysed.

Additional comments

In the abstract, the second sentence is about identifying people and the third sentence is about identifying perturbations. The articulation between the two is not clear.

The expression “an 11-person perturbative STS experiment” (p. 2 line 29 and p.3 line 21) is confusing. Please reformulate, for example : “a perturbative STS experiment with 11 subjects” or “11 subjects participated in a perturbative STS experiment”.

Switch in control strategy:

The authors first say a switch in control strategy is “indicated by a step or sit” (Abstract p.2 lines 30 -31), then p.9 line 35 “Previously in Sec. 2.1.3 we defined strategy switches as stepping or sitting back down”. However, in the Methods, the switch in control strategy is presented as an alternative to stepping or sitting: p.4 lines 5 – 6 “stepping, sitting back down, or switching control strategy”; p.4 lines 9 – 10 “take a step, sit back down, or switch control strategy”.

Moreover, in the rest of the text, control strategies actually refer to “natural”, “momentum transfer” and “quasi-static” (ex: p.4 lines 26 – 27).

This is very confusing.

2.1.1 Experimental protocol:

“Foot shift (FS) perturbations” (p.4, line 32): these are not really perturbed trials, so to avoid confusion they should perhaps simply be called “foot shift trials”. How many trials were performed at each foot position? Which strategy were the subjects instructed to use?

2.2.1 p.5 line 20: “We define the origin as the initial position of the ball of the subject’s foot”: is this indicated by a kinematic marker on a given anatomical landmark?

2.3 Computing Stability Basins

Definition 1: starting from the stability basin, the system will not necessarily arrive at the standing set, for example if the control inputs are not well chosen.

How many generators did you use for the standing set and the stability basins (i.e. what is the value of p)?

2.3.3 p. 9 lines 28 – 29 “CORA therefore assumes the input at a given time and state can take any value within the input bounds to create the SB.” How is this modified to calculate the Stability Basins for the LQR and FF+FB?

2.4.3 p.10 lines 25 – 26: In the leave-one-out procedure, why is the trial not also removed from the calculation of the standing set?

2.5.1 p.10 lines 40 to 42: the way in which the feedback matrix is determined is very obscure. What are the matrices Q and R and what does “best results” mean?

3. Results

p. 11 lines 34 – 35 “The subjects performed STS trajectories with generally the same characteristics as the (different) subjects examined by Shia”: what are these characteristics?

p.11 lines 41 – 42 “The SB shape and volume are highly dependent on the kinematic and temporal characteristics corresponding to the strategy selected, as in previous work”: how do the shape and volume change with strategy?

The units used to report distances are inconsistent: heights are reported in m, accelerations in m/s^2 , but foot positions in inches. Please consistently use the International System of Units and convert foot positions to m (or mm).

p.11 lines 55 – 56: how long in advance are failures predicted?

Figure 7: please indicate in the legend what the lines, boxes and whiskers correspond to (i.e. means, medians, percentiles?)

Figure 8: Please describe the figure in the main text. The light grey shading is too difficult to see. The names of the panels (a – Successful, b - Step, c – Sit) should be written horizontally. Please also illustrate a successful perturbation trial. The time during which the perturbation is active is indicated as a thick dashed line in Figure 1 and as a thick non-dashed line in Figure 8. Please choose a consistent representation.

Figure 9: the shades of blue and red are darker in the ground truth than in the other lines. The shades for incorrect predictions are too light.

Discussion

p. 15 line 2: the method nevertheless requires perturbation trials in order to determine the stability basin, which may not be convenient to administer in a clinical setting. Moreover, the stability basin does not indicate whether or not a subject will be able to successfully resist a given

perturbation. The classification of successful and unsuccessful perturbation trials relies on the motion capture data after the perturbation.

Appendix A

p. 18 lines 12 - 13: Does the range of perturbation onset timings cover the whole duration of the STS trials?

p. 18 lines 14 - 15: Is the baseline force level at the end of the perturbation non-zero? Is the baseline force level before the perturbation zero?

p. 18 line 16: Please report the range of peak forces across subjects, normalised by subject weight, for each of the three force levels. It would also be helpful to include a figure with the perturbation time course.

Appendix B p. 19 line 44: is the alignment done manually?

Author's Response to Decision Letter for (RSOS-191410.R0)

See Appendix A.

Decision letter (RSOS-191410.R1)

14-Nov-2019

Dear Mr Holmes,

On behalf of the Editors, I am pleased to inform you that your Manuscript RSOS-191410.R1 entitled "Characterizing the limits of human stability during motion: perturbative experiment validates a model-based approach for the Sit-to-Stand task" has been accepted for publication in Royal Society Open Science subject to minor revision in accordance with the Editor's suggestions. Please find the remaining comments at the end of this email.

Both the Associate and Subject Editor have recommended publication, but also suggest some minor revisions to your manuscript. Therefore, I invite you to respond to the comments and revise your manuscript.

- Ethics statement

- Data accessibility

It is a condition of publication that all supporting data are made available either as supplementary information or preferably in a suitable permanent repository. The data accessibility section should state where the article's supporting data can be accessed. This section should also include details, where possible of where to access other relevant research materials such as statistical tools, protocols, software etc can be accessed. If the data has been deposited in

an external repository this section should list the database, accession number and link to the DOI for all data from the article that has been made publicly available. Data sets that have been deposited in an external repository and have a DOI should also be appropriately cited in the manuscript and included in the reference list.

<http://datadryad.org/submit?journalID=RSOS&manu=RSOS-191410.R1>

- **Competing interests**

- **Authors' contributions**

- **Acknowledgements**

- **Funding statement**

Because the schedule for publication is very tight, it is a condition of publication that you submit the revised version of your manuscript before 23-Nov-2019. Please note that the revision deadline will expire at 00.00am on this date. If you do not think you will be able to meet this date please let me know immediately.

When submitting your revised manuscript, you will be able to respond to the comments made by the Editors and upload a file "Response to Referees" in "Section 6 - File Upload". You can use this to document any changes you make to the original manuscript. In order to expedite the processing of the revised manuscript, please be as specific as possible in your response to the remaining comments.

Kind regards,

on behalf of Dr Monica Daley (Associate Editor) and Kevin Padian (Subject Editor)
openscience@royalsociety.org

Associate Editor Comments to Author (Dr Monica Daley):

Thank you for the thorough and rigorous responses addressing the reviewer comments on your study of human stability during sit-to-stand tasks. I have looked over the response to the reviews and the revised manuscript, and I find the revised manuscript to be clear and thorough. I particularly appreciate the authors efforts to clearly present the procedures in Figure 6.

I have only a couple minor suggestions for final revisions to help readers who might be particularly interested in the experimental data. 1) It would be useful if the authors could provide more complete information about the number of individuals represented in each category of sit-to-stand strategy presented in Table 1 and other pooled statistical results. Or, put another way, how many trials were collected in each category for each individual? While I agree that the aggregated data is most informative for comparison to model predictions, it would be useful to know the extent to which individuals performed specific strategies idiosyncratically (or not). This information could be provided in a supplemental table if it is not easy to incorporate into the main paper. 2) Additionally, the authors may want to consider overlaying the data points onto the box plots in Figure 8, to provide better visual representation of the sample size and data distribution within each strategy.

Author's Response to Decision Letter for (RSOS-191410.R1)

See Appendix B.

Decision letter (RSOS-191410.R2)

26-Nov-2019

Dear Mr Holmes,

It is a pleasure to accept your manuscript entitled "Characterizing the limits of human stability during motion: perturbative experiment validates a model-based approach for the Sit-to-Stand task" in its current form for publication in Royal Society Open Science. The comments of the reviewer(s) who reviewed your manuscript are included at the foot of this letter.

on behalf of Dr Monica Daley (Associate Editor) and Kevin Padian (Subject Editor)
openscience@royalsociety.org

Appendix A

Characterizing the limits of human stability during motion: perturbative experiment validates a model-based approach for the Sit-to-Stand task Response to Reviewers

Patrick D. Holmes, Shannon M. Danforth, Xiao-Yu Fu, Talia Y. Moore, Ram Vasudevan

October 26, 2019

We thank the reviewers for their careful review of our paper. The provided comments were incredibly helpful and insightful. Based on the recommendations of the reviewers and editors, we have made several modifications to the paper, which are summarized in detail below. A modified and highlighted manuscript is attached to the end of this response letter. In the attached manuscript, new text to address comments is in blue, existing text that has been moved to address comments is in purple, removed text is in red and stricken out, and the corresponding reviewer’s comment is referenced in the format ‘reviewer#.comment’ in the margin. In the interest of keeping the response statement short, we have not copied, verbatim, the changes that are made in the paper. Figures and tables that appear only in the response letter are denoted with the prefix “R” before the number.

NOTE: major changes have been made to the manuscript that include:

- We have created a new subsection, Sec. 2.3, that explains and compares each of the proposed controllers to one other.
- We have renamed the BFF+FB controller as the “Input Bounds” controller, and give a more detailed interpretation of it in the manuscript.
- We have now wrapped the LQR controller training process in an optimization program that finds the Q and R that best fit the nominal, foot-shift, and successful perturbed trials to make the comparison more fair.
- We have renamed the standing set as the “target set” and clarified that it represents states that lead to standing, rather than states that describe standing itself.
- The appendix figure, table, and equation numbers now begin with the prefix A and start with 1.

Reviewer 1

Overall comments

This manuscript describes a study undertaken to assess the validity of basin of stability as obtained from a sit-to-stand experiment. To this aim, 11 subjects are asked to perform sit-to-stand movements, and perturbations are used. During some trials, subjects could not perform a successful sit-to-stand (due to perturbations). Results showed that basin of stability calculations could accurately predict which trials would be unsuccessful.

All in all, this is an interesting study, and the results seem promising. It is also written in a very readable way (except maybe the extensive use of abbreviations, which does not really help the reader). It could be because of my own limited background with the subject, but some of the methodological subtleties were less clear, and could thus be improved upon. I will try to outline these below.

Major comments

Comment 1 :

It is unclear to me how the BFF+FB controller differs from the other controllers, and I feel it would be good if the authors spend a few more words on this. For the LQR controller, it seems that one obvious difference is that it only uses non-perturbed trial data (am I correct here?). For the other controller, the difference is less clear to me. In order to appreciate the manuscript to its fullest, it would be good if the authors spend some more words on this.

We thank the reviewer for this helpful comment. We have updated Figure 3 to illustrate the LQR and FF+FB controllers in addition to the Input Bounds controller. Additionally, to clarify the differences between the controllers, we have consolidated the controller descriptions into a single subsection (Section 2.3), and added text comparing and contrasting the different controller models.

#1.1

Comment 2 :

Also, if my reading is correct and the BFF+FB controller used the perturbed trials (but only the parts which were still successful), while the LQR does not, what does that mean? Ideally, we would want to use these kind of measures to obtain a measure of stability from unperturbed movements, right?

We thank the reviewer for these insightful questions. The reviewer is correct that, ideally, we want to obtain measures of stability from unperturbed movements. The challenge is developing an accurate model for the controller from unperturbed movements, for which there is no consensus in the literature. In fact, we find that when all presented controllers are trained only on data from nominal and foot-shift trials, the resulting Stability Basins for the Input Bounds and FF+FB controllers have much lower accuracy. Table R1 below gives the classification rates of Stability Basins in which each controller is trained using nominal, foot-shift, and successful perturbed trials, compared to Stability Basins in which each controller is trained only using nominal and foot-shift trials. The target set generation and validation procedure for each are identical. The Stability Basins generated using the Input Bounds controller still have the highest rate of correct successful trial predictions, and the lowest rate of false step/sit predictions. However, the rate of correct successful trial predictions drops for both FF+FB and Input Bounds controllers. This is because these controllers are only trained on nominal and foot-shift trials. Therefore, we include successful perturbed trials when training all controllers to incorporate more of the successful control input space and form more accurate Stability Basins.

The reviewer is correct that the LQR controller in the previous version of our manuscript was trained on only unperturbed trials, while the Input Bounds controller was trained on unperturbed and successful perturbed trials. Because the feedback gains for the LQR controller are determined by minimizing a cost function specified by constant Q and R matrices, it was unclear to us how to incorporate data from successful perturbed trials to optimally choose these matrices. However, we have now wrapped the LQR controller training process in an optimization program that finds the Q and R that best fit the nominal, foot-shift, and successful perturbed trials. This new procedure is described in the manuscript in Section 2.3.1. We believe this process makes the comparison between the LQR controller and the other controllers more fair.

#1.2

Table R1: This table compares the results for Stability Basins when training on all nominal, foot-shift, and successful perturbed trials compared to training on only nominal and foot-shift trials. The results in the first two columns are the same results that appear in the paper (Table 4), and are presented here for reference. The accuracy of the Stability Basins for each method are compared. The results are aggregated across subjects and STS strategies. When trained only on nominal and foot-shift trials, the accuracy of the SBs formed using Input Bounds, LQR, and FF+FB controllers suffer.

Basin Type	trained on nominal, FS, successful perturbed		trained on nominal and FS	
	Successful Trials	Step/Sit Trials	Successful Trials	Step/Sit Trials
LQR	92/750 – 12.27%	197/198 – 99.49%	92/750 – 12.27%	196/198 – 98.99%
FF+FB	351/750 – 46.80%	195/198 – 98.48%	231/750 – 30.80%	196/198 – 98.99%
Input Bounds	718/750 – 95.73%	181/198 – 91.41%	501/750 – 66.80%	194/198 – 97.98%

Comment 3 :

Discussion, page 15, lines 6-7: “Furthermore, stability throughout a task can be studied by measuring the size and location of cross-sections of the SB at particular times.”; I wonder whether this would be informative? Does the size of the basin necessarily say how likely people are to get outside of the basin? I think most likely not?

We thank the reviewer for pointing out the lack of explanation for the utility of this analysis. We added references to our previous paper in which we used cross-sections of the SB at seat-off to show how the initial foot placement with respect to the center of mass affects the stability of the different control strategies in different ways.

#1.3

Minor comments

Comment 4 :

As there is no word limit, I suggest the authors remove most (if not all) abbreviations. These are mostly confusing for readers, especially for a broader audience (such as to be expected in this journal).

We thank the reviewer for their suggestion, and have removed most of the abbreviations from the text. In particular, we have renamed the BFF+FB controller to the “Input Bounds” controller. We have retained the abbreviation COM for Center of Mass, as this is commonly used in biomechanical studies.

#1.4

Comment 5 :

Abstract: “11 person perturbative STS experiment” sounds a bit odd? Consider changing to something like: Perturbation experiment, in which 11 subjects were included? (see also intro, page 2 line 21, and possibly other places)

We appreciate the reviewer’s suggestions for this phrase, which the other reviewer thought sounded odd as well. We have reworded this phrase in the abstract and elsewhere in the text.

#1.5

Comment 6 :

The definition of stability is only given in the methods, first few sentences. It might be good to have that earlier in the intro?

We thank the reviewer for their suggestion, and have moved the definition of stability to the introduction.

#1.6

Comment 7 :

Page 4, line 20: “We define the origin as the initial position of the ball of the subject’s foot”. However, in the figure, this appears to be the ankle? And how exactly is “the ball of the foot” defined?

We thank the reviewer for these clarifying questions. We have updated Figure 2 to display the proper location of the origin, and have added more detail about the precise definition of the origin to the manuscript. In particular, we have added the following sentence to the methods section: “Specifically, the origin is defined as the mean initial position of motion capture markers attached metatarsophalangeal joints on the subject’s left and right feet, as estimated by Visual3D.”

#1.7

Comment 8 :

Caption of figure 4: it may be good to change to “how X_t (the standing set) . . .”

We agree with the reviewer and have revised the manuscript accordingly. Additionally, we have renamed the standing set to the “target set” throughout, to more accurately represent that this set describes the set of states observed to lead to successful standing, rather than standing itself.

#1.8

Reviewer 2

Controller modelling assumptions

Comment 1 :

a) In Shia et al 2018, the control input is assumed to follow an open-loop control with feedback control around a nominal trajectory. This paper considers a more general control input, which is bounded for each time and state. From a biomechanical perspective, it makes sense that the ground reaction force is limited, and that the limits depend on the system state. However, I see no reason why the bounds should also depend on the normalised time, please justify this assumption.

b) The authors further assume that the bounds apply to the normalised control input, rather than the actual ground reaction force. I see however no reason why a subject should be able to produce larger forces in trials with a shorter duration.

c) Why are the bounds only fit on successful trials rather than on all trials? In unsuccessful trials, does the control input remain within the bounds determined from successful trials?

d) Why are the bounds (and the standing set) fit separately for Quasi-static, Natural, and Momentum transfer trials? I expect the same biomechanical constraints to apply for each of these tasks, resulting in the same bounds on the ground reaction force.

We appreciate the reviewer's thoughtful comments regarding the input bounds and have combined our responses to several comments together here for clarity.

a) The input bounds represent the range of inputs that are expected under a given strategy, rather than the range of all inputs that are biomechanically feasible. It is clear from the literature [1] that stability is strategy-dependent. Therefore, control inputs should be defined by the control strategy. If biomechanical limits are the only bounds in our models, we would have no ability to discern between the distinct control strategies that have been observed in our data and in previous research [1]. Rather, our bounds represent a range of uncertainty around performance of a given STS strategy. We appreciate that this was unclear in our first submission, so we have rewritten our explanation of input bounds in the manuscript with a motivating example described below:

Consider a scenario in which a person begins standing up using their natural STS strategy, but then stops halfway and remains in a crouched position. Certainly, the person can achieve this performance without violating biomechanical constraints on their joint torques or ground reaction forces. However, the person will have deviated from their natural STS strategy, in which they normally stand all the way up. Accordingly, the input that they applied would have exited the input bounds we have developed for their natural strategy at the time that they decided to remain in a crouched position. This is why the input bounds are trained on observations of STS rather than based on biomechanical constraints such as the size of the foot or joint torques. The input bounds represent the range of inputs that are expected under a given strategy, rather than the range of all inputs that are biomechanically feasible.

b) The reviewer is correct that the limits of the ground reaction force a subject can apply should not depend on trial duration. As mentioned above, the Input Bounds represent a range of inputs that are expected under a given strategy, rather than biomechanical constraints. The input scaling occurs because even within a given strategy, two trials may not necessarily take the same amount of time to complete.

#2.1

c) Because the Input Bounds represent a range of inputs that are expected under a given strategy, we want to ensure that the Input Bounds are trained only on that strategy. In the manuscript, we define a change in control strategy as a step or a sit; including these unsuccessful trials when training the Input Bounds would corrupt the training data.

The reviewer’s question about whether the control inputs during unsuccessful trials remain inside the bounds raises an interesting point. A trajectory’s input exceeding the input bounds is not a sufficient condition for the trajectory to exit a Stability Basin generated using Input Bounds. However, it is a necessary condition: if a trajectory exits a Stability Basin generated using the Input Bounds, then the trajectory’s input exceeds the Input Bounds at some point. Therefore, all unsuccessful trials that are correctly predicted to fail by the Input Bounds’ Stability Basins must have an input that exceeds the bounds at some point. We appreciate the reviewer’s interest in this point, and have added some text in Section 4 regarding necessary and sufficient conditions for exiting the Stability Basin to the manuscript’s discussion.

d) We fit separate bounds on each strategy to account for the dynamic differences observed during the performance of each strategy [1].

The reviewer’s question about the target set being fit separately for the different strategies is more subtle. During quiet standing, we expect that a person’s COM is positioned over their feet, that their COM has little to no horizontal velocity, and no vertical velocity [2]. While analyzing the data from our experiment, we found it difficult to quantify a consistent COM velocity threshold that defines when a subject had achieved quiet standing. This is because the velocities towards the end of the Quasi-Static strategy are significantly lower than the velocities towards the end of the Momentum-Transfer strategy.

Instead, the segmentation procedure we developed to determine the end of Sit-to-Stand focuses on final vertical displacement of the COM. This point occurs before the subject has reached quiet standing, but yields a more consistent segmentation of the trials. In particular, we define the end of the average nominal trials used for segmentation as when the vertical COM displacement exceeds 99% of its max. However, the final COM velocities at this point vary from strategy to strategy, as displayed in Fig. R1. This is why we fit strategy-specific target sets in the manuscript.

The target sets we define in the manuscript are better considered as “pre-standing” sets; they do not represent quiet standing itself, but rather the set of final trial states observed to lead to successful quiet standing for each Sit-to-Stand strategy. We have renamed the “standing set” in the previous manuscript as the “target set” in our revised manuscript to avoid confusion on this point.

#2.1

Comment 2 :

In equations (6) and (7), I think it is misleading to call the first term “feed-forward” and the second term “feed-back”. Presumably, after a perturbation the subjects will adjust the first term (within the bounds) to compensate for the perturbation, so it is not entirely feedforward.

We thank the reviewer for this insightful observation. As stated at the beginning of the response letter, one of the major changes we made in response to this comment is that we have renamed the “BFF+FB” controller as the “Input Bounds” controller to remove confusion regarding the relative contributions of feedforward versus feedback.

#2.2

Figure R1: The target sets, shown as the shaded regions, for subject ID 6’s Momentum-Transfer, natural, and Quasi-Static Sit-to-Stand strategies are shown. The final states (i.e. states at $t = 1$) of each of the subject’s nominal, foot-shift, and successful perturbed trials are shown as the colored dots for each strategy. The final states for each strategy are clustered in various regions of the state space, necessitating the target sets to be generated separately for each strategy.

Experimental comparison of the controllers

Comment 3 :

Why do the alternative controllers have such poor prediction rates for successful trials? If the stability basins for the LQR and FF+FB controllers are extended by 5% (similar to standing set), does this provide a better classification?

We thank the reviewer for these interesting questions. We have added some additional text to the manuscript’s discussion (Section 5) that further discusses the poor prediction rates of the FF+FB and LQR controllers for successful trials.

In response to the reviewer’s second question, we tested the accuracy of the Stability Basins generated by the LQR, FF+FB, and naive methods when they are dilated by 5%. The results and text that appear below now also appear within the results section (Section 3) of the manuscript:

The classification rates for these methods do improve, though not substantially. As seen in Table R2, the successful trial classification rate rises around 3% for the LQR controller, around 6% for the FF+FB controller, and around 5% for the naive method. To determine what would happen if we extended this analysis further, we also evaluated the Stability Basins when the basins are dilated by 25%. This further improves the successful trial classification rate of the LQR, FF+FB, and naive methods. However, at this dilation level, the FF+FB has a similar unsuccessful trial classification rate as the Input Bounds controller (89.41% compared to 91.41%), but a 41% higher false step/sit prediction rate (56.83% compared to 15.02%)..

This implies that the Input Bounds controller still yields the most accurate estimate of stability. The Input Bounds controller maintains a high accuracy for successful trials, while minimizing incorrect step and sit predictions. It requires no dilation to achieve this performance. The higher false step/sit prediction rates for the LQR and FF+FB controllers despite dilation indicates to us that the shape of the Stability Basin constructed using the Input Bounds controller is a better approximation of the actual bounds of performance.

#2.3

Table R2: The accuracy of the Stability Basins for each method are compared when dilating the LQR, FF+FB, and naive Stability Basins. The results are aggregated across subjects and STS strategies.

Basin Type	Successful Trials	Step/Sit Trials	False Successful Pred.	False Step/Sit Pred.
Input Bounds	718/750 – 95.73%	181/198 – 91.41%	17/735 – 2.31%	32/213 – 15.02%
LQR (no dilation)	92/750 – 12.27%	197/198 – 99.49%	1/93 – 1.08%	658/855 – 76.96%
FF+FB (no dilation)	351/750 – 46.80%	195/198 – 98.48%	3/354 – 0.85%	399/594 – 67.17%
naive (no dilation)	125/750 – 16.67%	195/198 – 98.48%	3/128 – 2.34%	625/820 – 76.21%
LQR (5% dilation)	113/750 – 15.07%	194/198 – 97.98%	4/117 – 3.42%	637/831 – 76.65%
FF+FB (5% dilation)	390/750 – 52.00%	192/198 – 96.97%	6/396 – 1.52%	360/552 – 65.22%
naive (5% dilation)	159/750 – 21.20%	195/198 – 98.48%	3/162 – 1.85%	591/786 – 75.19%
LQR (25% dilation)	160/750 – 21.33%	188/198 – 94.95%	10/170 – 5.88%	590/778 – 75.84%
FF+FB (25% dilation)	517/750 – 68.93%	177/198 – 89.39%	21/538 – 3.90%	233/410 – 56.83%
naive (25% dilation)	292/750 – 38.93%	189/198 – 95.45%	9/301 – 2.99%	458/647 – 70.79%

LQR controller

Comment 4 :

a) *In the Discussion, the authors mention that “the LQR controller proposed by Shia et al. [...] is not trained on any of the observed perturbed trials” (p. 14, lines 46 – 47). If this is also the case in your paper, then it is an unfair comparison.*

b) *In the Methods 2.5.1, the authors say that, to generate the feedback matrix for the LQR, they use “weighting matrices [which] were found empirically to produce the best results”. What does “best” mean? Does it provide the best classification performance for cable-pull perturbation trials?*

c) *2.5.1 p.10 lines 40 to 42: the way in which the feedback matrix is determined is very obscure. What are the matrices Q and R and what does “best results” mean?*

a) We thank the reviewer for their comment, and appreciate the reviewer’s concern. Reviewer 1 had similar concerns, which we have addressed in our response to their Comment 2, as well as in the text.

b) Previously the weighting matrices were manually chosen by iterating through combinations and finding the matrices that provided the best overall classification performance for both successful trials and step/sit trials.

c) However, to make this process more rigorous, we have adopted a new optimization based procedure for automatically choosing the weighting matrices, which we have detailed in our response to Comment 2 by Reviewer 1.

#2.4

FF+FB controller

Comment 5 :

Were the terms $ff(t)$ and $K(t)$ fit to the data from all successful trials, or only nominal and foot-shift trials? Are the values of $K(t)$ obtained from fitting the FF+FB controller similar to those obtained from fitting the BFF+FB controller?

We thank the reviewer for their questions. The terms $ff(t)$ and $K(t)$ of the FF+FB controller are fit to the data from all successful trials (except during the leave-one-out procedure), as is done for the Input Bounds controller. We have created Section 2.3 in the manuscript which defines each of the controllers, and 2.3.1 states that the Input Bounds, FF+FB, and now the LQR controller are all trained on the same subset of trials.

#2.5

The reviewer's second question reflects a close reading of the text; because the FF+FB and Input Bounds controllers are formed from a similar optimization procedure, we expect the values of $K(t)$ obtained for each to be similar. This is indeed the case. To quantify how close the values of $K(t)$ are for the FF+FB and Input Bounds controllers, we measured the angles and compared the magnitudes of the vectors comprising each $K(t)$.

First, let's write the gain matrix $K(t) \in \mathbb{R}^{2 \times 4}$ as the concatenation of $k_x(t), k_y(t) \in \mathbb{R}^4$ so that $K(t) = (k_x(t), k_y(t))^T$. The terms $k_x(t)$ and $k_y(t)$ represent gain vectors that map the state $x(t)$ to horizontal and vertical feedback inputs, respectively. Now, for a given subject and strategy, let $k_{x_1}(t)$ and $k_{y_1}(t)$ represent the gain vectors for the FF+FB controller, and let $k_{x_2}(t)$ and $k_{y_2}(t)$ represent the gain vectors for the Input Bounds controller. We can find the angle between these vectors, $\theta_x(t)$ and $\theta_y(t)$, by taking the inverse cosine of the normalized inner product of the vectors:

$$\theta_x(t) = \cos^{-1} \left(\frac{k_{x_1}(t) \cdot k_{x_2}(t)}{\|k_{x_1}(t)\| \|k_{x_2}(t)\|} \right), \quad \theta_y(t) = \cos^{-1} \left(\frac{k_{y_1}(t) \cdot k_{y_2}(t)}{\|k_{y_1}(t)\| \|k_{y_2}(t)\|} \right)$$

Each $\theta_x(t)$ and $\theta_y(t)$ may take a value in $[0, 180]$ degrees. After training each controller, we take the median value of $\theta_x(t)$ and $\theta_y(t)$ over 200 time steps to be a typical value for each, and define these as $\bar{\theta}_x$ and $\bar{\theta}_y$. Finally, by taking the mean and standard deviation of $\bar{\theta}_x$ and $\bar{\theta}_y$ across each of 33 comparisons (11 subjects \times 3 strategies), we find that

$$\begin{aligned} \bar{\theta}_x &\approx 15.31 \pm 5.16 \text{ degrees} \\ \bar{\theta}_y &\approx 5.20 \pm 1.95 \text{ degrees} \end{aligned}$$

Similarly, we can look at the ratio of the norms of the gain vectors, which we will refer to as $m_x(t)$ and $m_y(t)$, to see if the magnitudes of the vectors are similar:

$$m_x(t) = \frac{\|k_{x_1}(t)\|}{\|k_{x_2}(t)\|}, \quad m_y(t) = \frac{\|k_{y_1}(t)\|}{\|k_{y_2}(t)\|}$$

Again taking the median value of $m_x(t)$ and $m_y(t)$ over 200 time steps, then the mean and standard deviation across 33 comparisons, we find that

$$\begin{aligned} \bar{m}_x &\approx 1.20 \pm 0.14 \\ \bar{m}_y &\approx 1.04 \pm 0.08 \end{aligned}$$

The average values for $\bar{\theta}_x$ and $\bar{\theta}_y$ imply that in general, the angles between the gain vectors is small and therefore they have a similar direction. Similarly, the average values for $\bar{m}_x(t)$ and $\bar{m}_y(t)$ are close to 1, implying that the gain vectors share similar magnitude. These results imply that overall, the values of $K(t)$ obtained from fitting the FF+FB controller and the Input Bounds controller are similar, though not identical.

#2.5

BFF+FB controller

Comment 6 :

If the stability basins are calculated only from nominal and foot-shift trials, does this accurately predict success and failure in cable-pull perturbation trials?

We thank the reviewer for their question, and respectfully refer them to our response to Reviewer 1's Comment 2.

#2.6

Organisation of the Methods

Comment 7 :

All of the information relevant to the experiment and its analysis should be in a single section (presumably 2.1 Perturbative STS experiment):

The following information belongs to the Methods rather than the Results:

1. P. 11 lines 32 – 34 “We collected 948 trials from 11 participants (three female and eight male; ages 18-32; height 1.70 +/- 0.12 m; body mass 65.4 +/- 10.2 kg), including 213 unperturbed, 194 foot-shifted, and 591 cable pull trials. Each subject gave their informed written consent, and had no physical or balance disorders which could affect their ability to perform STS.”
2. p. 11 lines 51 – 52 “SBs were generated on a laptop computer with a 2.7 GHz Intel Core i7 processor. SBs computed using the BFF+FB controller take 0:76+/-0:014 seconds to compute.”

We appreciate the reviewer’s perspective and have reorganized accordingly.

#2.7

Comment 8 :

2.1.3 Classifying trials as successful/unsuccessful (p.5 lines 2 to 8):

It is inconvenient for stepping and sitting onset to be defined several pages later (p.9 lines 41 to 47); the definition should be moved to Section 2.1.3. The classification of foot-shift trials (found p.11 lines 43 to 46) should also be moved here, and it should be clarified: when a subject is unable to initiate the STS movement, or violates the parameters of acceptable trials, then the trial should not be considered successful. Please replace “this scheme can classify” (p.5 line 7) by “this scheme classifies”.

We thank the reviewer for this suggestion and have reorganized accordingly.

#2.8

Comment 9 :

The three different controllers (BFF+FB, LQR, FF+FB) should be presented in a single section, to highlight their differences. Moreover, the modelling assumptions for each controller need to be justified.

We appreciate the reviewer’s suggestion and have drastically reorganized the manuscript accordingly. We agree that this organization method is more general and logical, and have therefore created the new Section 2.3 to describe and compare all controller models.

#2.9

Comment 10 :

Figure 3 should be extended to illustrate the LQR and FF+FB controllers, similar to panel (b). It would be helpful to color-code the dots in panel (b) to show those that correspond to nominal, foot-shift, successful perturbation and unsuccessful perturbation trials. It would also be useful to indicate which of these trials were used to fit the three different controllers (as it seems the controllers were not all fit on the same subset of trials).

We thank the reviewer for this helpful suggestion, and have updated Figure 3 to illustrate the LQR and FF+FB controllers. We have color coded the dots in the panels to correspond to nominal, foot-shift, and successful perturbation trials. However, we have left the unsuccessful perturbation trials out of this figure, because these trials are not used for training the controllers. We have updated the manuscript (Section 2.3.1) to clarify that all controllers are now trained on the same subset of trials, whereas in our prior submission the LQR controller was not trained on the successful perturbation trials.

#2.10

Comment 11 :

The procedure for computing stability basins should be briefly described in comprehensible terms and explained through a sketch, but most of the mathematical definitions as well as section 2.4.2 should be moved to the Appendix.

We have considered the reviewer’s suggestion, but it is our opinion that this section is a key contribution of this manuscript distinct from previous work and essential to replication of our methods. We appreciate the reviewer’s perspective, and are including an animated GIF as a supplemental figure to graphically depict the construction of the stability basin.

#2.11

Comment 12 :

The procedure for validating the stability basins is difficult to follow. Could you please provide a graphical representation indicating which trials are used for which steps of the procedure, for each of the strategies and the controllers? For example, it is not clear to me in which step the foot-shift trials are analysed.

We thank the reviewer for this suggestion. We have created the new Figure 6 to clearly sketch out the Stability Basin validation procedure.

#2.12

Additional comments

Comment 13 :

In the abstract, the second sentence is about identifying people and the third sentence is about identifying perturbations. The articulation between the two is not clear.

We agree with the reviewer, and have revised this portion of the abstract accordingly.

#2.13

Comment 14 :

The expression “an 11-person perturbative STS experiment” (p. 2 line 29 and p.3 line 21) is confusing. Please reformulate, for example : “a perturbative STS experiment with 11 subjects” or “11 subjects participated in a perturbative STS experiment”.

This was also mentioned in Comment 5 by Reviewer 1, so we have reworded this phrase in the abstract and elsewhere in the text.

#2.14

Comment 15 :

Switch in control strategy:

The authors first say a switch in control strategy is “indicated by a step or sit” (Abstract p.2 lines 30 -31), then p.9 line 35 “Previously in Sec. 2.1.3 we defined strategy switches as stepping or sitting back down”. However, in the Methods, the switch in control strategy is presented as an alternative to stepping or sitting: p.4 lines 5 – 6 “stepping, sitting back down, or switching control strategy”; p.4 lines 9 – 10 “take a step, sit back down, or switch control strategy”. Moreover, in the rest of the text, control strategies actually refer to “natural”, “momentum transfer” and “quasi-static” (ex: p.4 lines 26 – 27). This is very confusing.

We thank the reviewer for pointing out this ambiguity in the text. We have revised the text to indicate that stepping or sitting represents the “failure of a control strategy.”

#2.15

Comment 16 :

2.1.1 Experimental protocol:

“Foot shift (FS) perturbations” (p.4, line 32): these are not really perturbed trials, so to avoid confusion they should perhaps simply be called “foot shift trials”. How many trials were performed at each foot position? Which strategy were the subjects instructed to use?

We thank the reviewer for these questions. We have clarified these details in the “Experimental protocol” section of the manuscript.

#2.16

Comment 17 :

2.2.1 p.5 line 20: “We define the origin as the initial position of the ball of the subject’s foot”: is this indicated by a kinematic marker on a given anatomical landmark?

We thank the reviewer for this clarifying question, and refer them to our response to Comment 7 by Reviewer 1.

#2.17

Comment 18 :

2.3 Computing Stability Basins

Definition 1: starting from the stability basin, the system will not necessarily arrive at the standing set, for example if the control inputs are not well chosen.

We would point out to the reviewer that in Definition 1 (now Definition 2), we specify that the trajectory must obey the control model. This guarantees that a trajectory starting within the Stability Basin will arrive at the target set.

#2.18

Comment 19 :

2.3 Computing Stability Basins

How many generators did you use for the standing set and the stability basins (i.e. what is the value of p)?

We appreciate the reviewer’s question, and have added a sentence to the text explaining this. Four generators are utilized for each target set, and while the precise number of generators used for each zonotope comprising the Stability Basin is decided by the CORA toolbox, we have set the maximum number of generators to be 800.

#2.19

Comment 20 :

2.3.3 p. 9 lines 28 – 29: “CORA therefore assumes the input at a given time and state can take any value within the input bounds to create the SB.” How is this modified to calculate the Stability Basins for the LQR and FF+FB?

We thank the reviewer for this clarifying question. The inputs to CORA at each time and state may either be a point (as in the FF+FB and LQR controllers) or a set (as in the Input Bounds controller). We have now revised the manuscript so that Sec 2.3 introducing each controller model comes before the Stability Basin computation section, and have made this point more clear in the text.

#2.20

Comment 21 :

2.4.3 p.10 lines 25 – 26: *In the leave-one-out procedure, why is the trial not also removed from the calculation of the standing set?*

We appreciate the reviewer’s question and have added to the manuscript to reflect the following response. The target set for each subject and Sit-to-Stand strategy is generated by essentially taking the convex hull of a set of 4-dimensional points corresponding to the final states of all nominal, foot-shift, and successful perturbed trials. (In reality, we are actually wrapping this convex hull in a zonotope with 4 generators as in Fig. R1, but for the purposes of this discussion assume it is a convex hull.) If we follow a leave-one-out procedure when generating this convex hull, then the points that are on the boundary of the original convex hull are necessarily outside of the new convex hull when they are left out.

#2.21

This means that successful trials whose final states lie on the edge of the original target set necessarily exit the Stability Basins, and therefore are automatically misclassified when following this leave-one-out procedure. Indeed, as shown in Table R3, this drops the successful trial classification rate of Stability Basins for all controller models. The classification rate for the naive method remains the same, since it does not rely on the target set for generating a Stability Basin. The reduction in accuracy of the controller models' Stability Basins arises due to the low number of trials collected per subject. Larger sample sizes should decrease the effect of missing data on the target set generation.

For these reasons, this procedure serves to validate the robustness of the target set generation, rather than the Stability Basins themselves, because the drop in classification rates is directly attributable to whether or not the trajectories' final states lie on the boundary of the original target set. Note also that simply identifying the trials that terminate outside of the target set can only detect failure after a movement has been completed. By using the target set, controller models, and reachability analysis, the Stability Basin can be used to detect failure before the failure actually occurs. Thus, leaving a trial out of the controller training and not the target set is a more fair test of the Stability Basins' predictions of failure.

#2.21

Table R3: When a leave-one-out procedure is used to generate the target sets for testing successful trials, the accuracy of the SBs formed using Input Bounds, LQR, and FF+FB controllers suffer. In particular, the rate of false step/sit predictions increases for each. The results of the comparison are reported below, and are aggregated across subjects and STS strategies.

Basin Type	Successful Trials	Step/Sit Trials	False Successful Predictions	False Step/Sit Predictions
Input Bounds	572/750 – 76.05%	191/198 – 96.46%	17/589 – 2.89%	178/359 – 49.58%
LQR	75/750 – 10.00%	197/198 – 99.49%	1/76 – 1.32%	675/872 – 77.41%
FF+FB	284/750 – 37.87%	195/198 – 98.48%	3/287 – 1.05%	466/661 – 70.50%
naive	125/750 – 16.67%	195/198 – 98.48%	3/128 – 2.34%	625/820 – 76.21%

Comment 22 :

3. Results, p. 11 lines 34 – 35: “The subjects performed STS trajectories with generally the same characteristics as the (different) subjects examined by Shia”: what are these characteristics?

We thank the reviewer for pointing out this ambiguous description. We have clarified the text to indicate that the subjects from both studies exhibited similar maximum and minimum accelerations for each control strategy.

#2.22

Comment 23 :

3. Results, p. 11 lines 41 – 42: “The SB shape and volume are highly dependent on the kinematic and temporal characteristics corresponding to the strategy selected, as in previous work”: how do the shape and volume change with strategy?

We respectfully direct the reviewer to our response to Comment 1 by Reviewer 1.

#2.23

Comment 24 :

The units used to report distances are inconsistent: heights are reported in m, accelerations in m/s^2 , but foot positions in inches. Please consistently use the International System of Units and convert foot positions to m (or mm)

We thank the reviewer for this suggestion, and have implemented these changes.

#2.24

Comment 25 :

p.11 lines 55 – 56: how long in advance are failures predicted?

We thank the reviewer for their question, and have included these results in Table R4. Though the Stability Basins generated by the LQR, FF+FB, and naive methods predict failure earlier than the Input Bounds method, they generate false step/sit predictions upwards of 67% of the time (as shown in Table R1), making their predictions untrustworthy.

#2.25

Table R4: This table shows how long in advance failures are predicted for each of the Stability Basins. The average time is given in terms of normalized time (e.g. $t = 0.5$ is equivalent to 50% of a Sit-to-Stand trial). Note that these results are only for Stability Basin predictions of failure that were actually correct.

Basin Type	Prediction to Step	Prediction to Sit
Input Bounds	0.28	0.19
LQR	0.48	0.37
FF+FB	0.41	0.30
naive	0.46	0.34

Comment 26 :

Figure 7: please indicate in the legend what the lines, boxes and whiskers correspond to (i.e. means, medians, percentiles?)

We appreciate the reviewer’s suggestion, and have added this information to the figure’s caption.

#2.26

Comment 27 :

Figure 8: Please describe the figure in the main text. The light grey shading is too difficult to see. The names of the panels (a – Successful, b - Step, c – Sit) should be written horizontally. Please also illustrate a successful perturbation trial. The time during which the perturbation is active is indicated as a thick dashed line in Figure 1 and as a thick non-dashed line in Figure 8. Please choose a consistent representation.

We have incorporated the reviewer’s helpful suggestions into the design of the new Figure 6.

#2.27

Comment 28 :

Figure 9: the shades of blue and red are darker in the ground truth than in the other lines. The shades for incorrect predictions are too light.

We thank the reviewer for the feedback. The color transparencies are now consistent across all rows and the shades for incorrect predictions are slightly darker.

#2.28

Comment 29 :

a) Discussion, p. 15 line 2: the method nevertheless requires perturbation trials in order to determine the stability basin, which may not be convenient to administer in a clinical setting.

b) Moreover, the stability basin does not indicate whether or not a subject will be able to successfully resist a given perturbation.

c) The classification of successful and unsuccessful perturbation trials relies on the motion capture data after the perturbation.

We thank the reviewer for their nuanced comments.

a) The reviewer is correct that the method presented in the manuscript relies on data from successful perturbed trials to construct the Stability Basins. However, we would like to point to our response to Reviewer 1's Comment 2 and Table R1. There, we show that the Input Bounds controller model is still able to correctly classify 66.8% of successful trials when trained only on unperturbed data. Granted, this is a 29% drop in accuracy compared to when the Input Bounds are trained on successful perturbed trials as well, but we are training on less than half as many trials (357 total nominal and foot-shift trials, 393 successful perturbed trials). Note that the accuracy of the FF+FB controller model drops as well, and that the Input Bounds model still classifies successful trials with 36% greater accuracy. We hypothesize that with additional unperturbed data for each subject, the accuracy of the Stability Basins formed only from unperturbed data will improve.

b) We respectfully disagree with the suggestion that the Stability Basin does not indicate whether or not a subject will be able to successfully resist a given perturbation.

First, we have learned from our experiment that knowing the magnitude and direction of a given perturbation is not sufficient to predict the subject's response. In fact, the subject's state, Sit-to-Stand control strategy, and time of perturbation application are all key features that determine the subject's response to perturbation. All of these factors are incorporated into the Stability Basin's predictions of whether or not the subject can resist a given perturbation.

Next, let us consider the simplest perturbation possible to our model, which are horizontal and vertical impulses applied to the COM. These perturbations instantaneously change the horizontal and vertical velocities of the COM. Given the time of perturbation application, we can examine the cross section of the subject's Stability Basin at that time. Starting at the subject's current COM state, the changes in horizontal and vertical and vertical velocities needed to reach the boundary of the Stability Basin cross section represent the set of impulses that the subject can successfully resist.

In reality, such a perfect COM perturbation does not exist. However, if the effect of a perturbation on the subject's COM can be modeled, then the Stability Basin should predict whether or not the subject can resist it.

c) We interpret this comment to question whether the Stability Basins' predictions can be used to predict failure in real-time (i.e., do not require post-perturbation data to predict failure).

Once a Stability Basin has been constructed, trajectories of the COM can be tested to see if they exit the Stability Basin in real-time. Failures can therefore be predicted before failure occurs, as is demonstrated in Table R4. Any device that estimates the state of the subject's COM in real-time could be used in conjunction with the Stability Basins to predict failure of a Sit-to-Stand strategy online.

#2.29

Comment 30 :

a) *Appendix A, p. 18 lines 12 - 13: Does the range of perturbation onset timings cover the whole duration of the STS trials?*

b) *Appendix A, p. 18 lines 14 - 15: Is the baseline force level at the end of the perturbation non-zero? Is the baseline force level before the perturbation zero?*

a) We thank the reviewer for their question. The perturbations did not necessarily cover the entire duration of the STS trials, but were generally focused around seatoff. Perturbations were usually applied around 50% of the way through the motion (shown in the Pert. Onset column of Table 1 of the manuscript).

b) We have added additional details to the text regarding the baseline tensioning force, as well as Figure A1 displaying a typical perturbation time course. As appears verbatim in the paper: "...both the front and back perturbation motors were commanded to maintain a baseline tensioning force of 19.6 N, so that the net force on the subject was approximately 0 N when no active perturbation was applied."

#2.30

Comment 31 :

Appendix A, p. 18 line 16: Please report the range of peak forces across subjects, normalised by subject weight, for each of the three force levels. It would also be helpful to include a figure with the perturbation time course

We thank the reviewer for this useful suggestion. We have included Appendix Figure A2 of the manuscript reporting the peak forces across subjects, normalised by weight, for each force level, as well as a figure showing a typical perturbation time series.

#2.31

Comment 32 :

Appendix B p. 19 line 44: is the alignment done manually?

The alignment for Natural and Momentum-Transfer trials is done automatically, while alignment for Quasi-Static trials is done manually. We thank the author for pointing out this ambiguity, and have clarified this point in the text.

#2.32

References

- [1] Rachid Aissaoui and J Dansereau. Biomechanical analysis and modelling of sit to stand task: a literature review. In IEEE Conference on Systems, Man, and Cybernetics, volume 1, pages 141 – 146 vol.1, 02 1999.
- [2] David A Winter, Aftab E Patla, Milad Ishac, and William H Gage. Motor mechanisms of balance during quiet standing. Journal of Electromyography and Kinesiology, 13(1):49 – 56, 2003.

Appendix B

COLLEGE OF ENGINEERING
MECHANICAL ENGINEERING
UNIVERSITY OF MICHIGAN
G058 Automotive Laboratory
1231 Beal Avenue
Ann Arbor, MI 48109-2125
734 647 5560

November 21st, 2019

Dear Editorial Board,

Enclosed, please find our revised manuscript, "*Characterizing the limits of human stability during motion: perturbative experiment validates a model-based approach for the Sit-to-Stand task*" by Patrick D. Holmes, Shannon M. Danforth, Xiao-Yu Fu, Talia Y. Moore, and Ram Vasudevan.

We thank you for the opportunity to publish in Royal Society Open Science. In particular, we thank Dr. Monica Daley for overseeing this paper through the review process. We have made the following edits to our submission to incorporate Dr. Daley's useful suggestions:

- We have created a supplementary material document which gives experimental statistics and Stability Basin results for individual subjects and Sit-to-Stand strategies. We have pointed to the supplementary material in the appropriate table captions of the manuscript.
- We have overlaid data points onto the box plots in Figure 8 to better show the sample size and distribution of the data.
- We have also made small grammatical and notational edits to improve readability of the manuscript.

We hope that you enjoy reading the updated and improved manuscript. We thank you for your time and effort throughout the revision process.

Sincerely,

Ram Vasudevan
Assistant Professor
Mechanical Engineering
University of Michigan